**Sources and sinks driving sulphuric acid concentrations in contrasting environments:**
**implications on proxy calculations**
Lubna Dada[1,2], Ilona Ylivinkka[2,3], Rima Baalbaki[2], Chang Li[1], Yishuo Guo[1], Chao Yan[1,2], Lei Yao[1,2],
Nina Sarnela[2], Tuija Jokinen[2], Kaspar R. Daellenbach[2], Rujing Yin[4], Chenjuan Deng[4], Biwu Chu[1,2],
Tuomo Nieminen[2], Yonghong Wang[1,2], Zhuohui Lin[1], Roseline C. Thakur[2], Jenni Kontkanen[2],
Dominik Stolzenburg[2], Mikko Sipilä[2], Tareq Hussein[2], Pauli Paasonen[2], Federico Bianchi[2], Imre
Salma[5], Tamás Weidinger[6], Michael Pikridas[7], Jean Sciare[7], Jingkun Jiang[4], Yongchun Liu[1], Tuukka
Petäjä[2], Veli-Matti Kerminen[2], Markku Kulmala[1,2]
[1] Aerosol and Haze Laboratory, Beijing Advanced Innovation Center for Soft Matter Science and Engineering, Beijing
University of Chemical Technology, Beijing, China.
[2] Institute for Atmospheric and Earth System Research INAR / Physics, Faculty of Science, University of Helsinki,
Finland
[3] SMEAR II station, University of Helsinki, 35500 Korkeakoski, Finland
[4] State Key Joint Laboratory of Environment Simulation and Pollution Control, School of Environment, Tsinghua
University, 100084 Beijing
[5] Institute of Chemistry, Eötvös University, 1518 Budapest, P.O. Box 32, Hungary
[6] Department of Meteorology, Eötvös University, H-1518 Budapest, P.O. Box 32, Hungary
[7] The Cyprus Institute, Climate & Atmosphere Research Centre (CARE-C), 20 Konstantinou Kavafi Street, 2121, Nicosia,
Cyprus
*Correspondence to: markku.kulmala@helsinki.fi*
**Abstract**
Sulphuric acid has been shown to be a key driver for new particle formation and subsequent growth
in various environments mainly due to its low volatility. However, direct measurements of gas-phase
sulphuric acid are oftentimes not available, and the current sulphuric acid proxies cannot predict for
example its nighttime concentrations or result in significant discrepancies with measured values.
Here, we define the sources and sinks of sulphuric acid in different environments and derive a new
physical proxy for sulphuric acid to be utilized in locations and during periods when it is not
measured. We used $H_2SO_4$ measurements from four different locations: Hyytiälä, Finland; Agia
Marina, Cyprus; Budapest, Hungary; and Beijing, China, representing semi-pristine boreal forest,
rural environment in the Mediterranean area, urban environment and heavily polluted megacity,
respectively. The new proxy takes into account the formation of sulphuric acid from $SO_2$ via OH
oxidation and other oxidation pathways, specifically that via stabilized Criegee Intermediates. The
sulphuric acid sinks included in the proxy are its condensation sink (CS) and atmospheric clustering
starting from $H_2SO_4$ dimer formation. Indeed, we found that the observed sulphuric acid
concentration can be explained by the proposed sources and sinks with similar coefficients in the four
contrasting environments where we have tested it. Thus, the new proxy is a more flexible and an
important improvement over previous proxies. Following the recommendations in the manuscript, a
proxy for a specific location can be derived.

Keywords: sulphuric acid, proxy, boreal, rural, urban, megacity

## 1. Introduction

Atmospheric New Particle formation (NPF) events and their subsequent growth have been observed to take place almost everywhere in the world (Kulmala et al., 2004; Kerminen et al., 2018). Many of these observations are based on continuous measurements and some include more than a year of measurement data (Nieminen et al., 2018). The importance of NPF events on the global aerosol budget and cloud condensation nuclei formation has been well established (Spracklen et al., 2008; Merikanto et al., 2009; Spracklen et al., 2010; Kerminen et al., 2012; Gordon et al., 2017). Recently, the contribution of NPF to haze formation, which was still controversial, is being investigated in an increasing number of studies from Chinese megacities (Guo et al., 2014).

Sulphuric acid ($H_2SO_4$), which has a very low saturation vapor pressure and strong hydrogen bonding capability (Zhang et al., 2011), has been found to be the major precursor of atmospheric NPF (Weber et al., 1996; Kulmala et al., 2004; Sihto et al., 2006; Sipilä et al., 2010; Erupe et al., 2011; Lehtipalo et al., 2018; Ma et al., 2019) and is often used in global models for simulating the occurrence and intensity of new particle formation events (Dunne et al., 2016). However, atmospheric measurements of gas-phase sulphuric acid are rare, mainly due to its low concentration ($10^6$–$10^7$ molecules cm$^{-3}$ or below) that can only be measured using state-of-the art instruments (Mikkonen et al., 2011) such as the Chemical Ionization atmospheric pressure interface time of flight spectrometer (CI-APi-ToF) (Eisele and Tanner, 1993; Jokinen et al., 2012). Therefore, a physically and chemically sound proxy is needed to estimate $H_2SO_4$ concentrations in various environments where NPF events are observed but $H_2SO_4$ concentrations are not continuously measured.

Due to its important participation in clustering and thus in the NPF process, several studies have tried to produce proxies for $H_2SO_4$ in order to fill gaps in data. For example, Petäjä et al. (2009) developed an approximation of gas-phase $H_2SO_4$ concentration in Hyytiälä, southern Finland, using its source from reactions between $SO_2$ and OH radicals, and its loss by condensation onto pre-existing particles (condensation sink, CS). Later, Mikkonen et al. (2011) developed $H_2SO_4$ proxies based on measurements at six urban, rural and forest areas in European and North American sites. Proxies developed by Mikkonen et al. (2011) suggested that the sulphuric acid concentration depends mostly on the available radiation and $SO_2$ concentration, with little influence by CS. However, Lu et al. (2019), who developed a daytime proxy based on measurement in Beijing China, suggested the need of taking into account the CS when approximating gaseous $H_2SO_4$, especially in areas where the condensational sink can be relatively high. The proxy developed by Lu et al. (2019) takes into consideration the formation pathways of $H_2SO_4$ via OH radicals from both the conventional photolysis of $O_3$ and from the photolysis of HONO, as well as, the loss of $H_2SO_4$ via CS. Besides the previously-developed proxies, an additional proxy is still needed for representing nighttime periods which were not considered previously.

Here, we derive a new proxy which takes into account the production of gaseous sulphuric acid from $SO_2$ with oxidation by OH and stabilized Criegee Intermediates (Mauldin et al., 2012) reactions, and its losses onto pre-existing aerosol particles (condensation sink) and due to molecular cluster formation. In order to evaluate our hypothesized sources and sinks and derive the proxy equations, we utilize measurements from four different locations: (1) Hyytiälä, Finland, (2) Agia Marina, Cyprus, (3) Budapest, Hungary and (4) Beijing, China, representing a semi-pristine boreal forest environment, rural environment in the Mediterranean area, urban environment and heavily polluted

megacity, respectively. To evaluate the predictive power of the derived proxies, the equations are further tested on independent data sets. We further compare the coefficients of production and losses in each environment in order to understand the prevailing mechanism of the $H_2SO_4$ budget in each of the studied environments. As a result of this investigation, a well-defined sulphuric acid concentration can be derived for multiple areas around the world and even extended in time during times when it was not measured (such as: gap filling, forecast, prediction, estimation, etc.).

## 2. Measurement locations, observations and instrumentation

### 2.1. Locations
**Semi-pristine boreal forest environment: Hyytiälä, Finland**

Measurements were conducted at the SMEAR II-station (Station for Measuring Ecosystem–Atmosphere Relations), located in Hyytiälä (61.1° N, 24.17°E, 181 m a.s.l. (Hari and Kulmala, 2005)), southern Finland. Here we used measurements from August 18, 2016 to June 5, 2017 and from March 8, 2018 to February 28, 2019. The data from 2016, 2018 and 2019 was used as a training data set for developing the proxy equation, while the data from 2017 was used for testing the predictive power of the developed proxy. A summary for all locations and instrumentation is given in Tables S1 (training data sets) and S2 (testing data sets).

**Rural background site: Agia Marina, Cyprus**

Measurements were conducted at the Cyprus Atmospheric Observatory (CAO) (35.03° N, 33.05° E; 532 m a.s.l.), a rural background site located close to Agia Marina Xyliatou village, between February 22 and March 3, 2018. For more details, see for example Pikridas et al. (2018). The data set from this location is used solely as a training data set.

**Semi-urban site: Helsinki, Finland**

Measurements were conducted at the SMEAR III-station, located in Helsinki (60.20° N, 24.96° E, 25 m a.s.l.). For more details about the location see for example Hussein et al. (2008). Here, we measured from July 1, 2019 to July 16 2019 as a testing data set.

**Urban location: Budapest, Hungary**

The measurements took place at the Budapest platform for Aerosol Research Training (BpART) Research Laboratory (47.47° N, 19.06° E, 115 m a.s.l.) of the Eötvös University situated on the bank of the Danube between March 21 and April 17, 2018. The site represents a well-mixed average atmosphere of the city centre Salma et al. (2016a). The data set from this location is used solely as a training data set.

**Polluted megacity: Beijing, China**

Here, observations performed at the west campus of Beijing University of Chemical Technology (39.94° N, 116.30° E) between March 15, 2019 and June 15, 2019 were used as a training data set

while observations from September 8, 2019 to October 15, 2019 where used as a testing data set. The
sampling took place from outside the window at the 5[th] floor of a university building adjacent to a
busy street. For more details, see for example Lu et al. (2019); Zhou et al. (2020).
**Near an oil-refinery industrial area:  Kilpilahti, Finland**
The measurement took place at Nyby measurement station (60.31° N, 25.50° E) between June 07 and
June 29, 2012. The site is within 1.5 km close to Neste Oy. oil refinery and Kilpilahti industrial area.
For more information on the site, please see Sarnela et al. (2015). The data set from this location is
used solely as a testing data set.
2.2. **Instrumentation**
**Trace Gases**
A summary for all locations and instrumentation is given in Tables S1and S2. Measurements of
different variables within the same location are performed at the same platform unless specified
otherwise. In all locations, the sulphuric acid concentrations were measured using a Chemical
Ionization atmospheric pressure interface time of flight spectrometer (CI-APi-ToF) (Eisele and
Tanner, 1993; Jokinen et al., 2012) with $NO_3^-$ as a reagent ion and analyzed using a tofTools package
based on MATLAB software (Junninen et al., 2010). In all locations, the CI-APi-ToF instruments
were calibrated in a similar way prior to the campaign using the method presented by Kurten et al.
(2012) to ensure the results from different sites are comparable. In Hyytiälä, the sulphuric acid
concentrations were measured at the tower 35 m above ground level. In Helsinki, the sulphuric acid
concentrations were measured from the 4[th] floor window (~12 m above ground level) of the university
building adjacent (~200 m) to the SMEAR III station. In Hyytiälä, and Beijing, the $SO_2$ and $O_3$
concentrations were measured using an $SO_2$ analyzer (Model 43i, Thermo, USA), with a detection
limit of 0.1 ppbv, and $O_3$ analyzer (Model 49i, Thermo, USA), respectively. In Hyytiälä, the trace
gases concentrations were measured at the tower 16.8 m above ground level. In Helsinki, the $SO_2$
concentrations were monitored at a 32 m tower at the SMEAR III station using UV-fluorescence
(Horiba APSA 360). In Agia Marina, $SO_2$ and $O_3$ are monitored using Ecotech Instruments (9850 and
9810, respectively). Concentrations of $SO_2$ in Budapest were measured by UV fluorescence
(Ysselbach 43C) with a time resolution of 1 h at a station of the National Air Quality Network located
1.7 km in the upwind prevailing direction from the BpART site. It was shown earlier that the hourly
average $SO_2$ concentrations (See Figure S1) in central Budapest are ordinarily distributed without
large spatial gradients (Salma and Németh, 2019; Mikkonen et al., 2020). In Kilpilahti, $SO_2$
concentration were measured using Thermo Scientific ™ Model 43i $SO_2$ Analyser at Neste Oil
refinery. Trace gases measured during the short campaign periods in Agia Marina, and Budapest are
representative of yearly concentrations in respective locations when compared to longer term
measurements at the same site (Salma et al., 2016b; Baalbaki, 2020, In Prep.; Mikkonen et al., 2020).
**Particle number Size Distribution**
The condensation sink (CS) was calculated using the method proposed by Kulmala et al. (2012) from
number size distribution measurements. In Hyytiälä, the particle number size distribution was
measured using a twin differential mobility particle sizer (DMPS) (Aalto et al., 2001). In Agia Marina,

the particle number size distribution between 2 and 800 nm was reconstructed from two instruments: an Airel NAIS (Neutral cluster and Air Ion Spectrometer, 2-20 nm) and TSI SMPS (Scanning Mobility Particle Sizer, 20-800 nm). In Helsinki, a twin-DMPS system (diameter 3–950 nm) was used to monitor the particle number size distribution. In Budapest, the particle number size distribution was measured by a flow-switching type DMPS in a diameter range from 6 to 1000 nm in the dry state of particles (RH<30%) in 30 channels with a time resolution of 8 min (Salma et al., 2016a). In Beijing, the particle number size distribution between 3 nm and 850 nm was measured using a Particle Size Distribution System (PSD,(Liu et al., 2016)). Condensation sink obtained at Kilpilahti was acquired from particle number size distribution measured using a DMPS (6- 1000 nm). Although having a diurnal cycle, condensation sink values obtained during the short campaign periods in Agia Marina and Budapest are representative of yearly concentrations in respective locations when compared to longer term measurements at the same site (Salma et al., 2016b; Baalbaki, 2020, In Prep.).

## Radiation

In Hyytiälä, Global radiation (GlobRad) was measured using a SK08 solar pyranometer until August 24, 2017 and after that using a EQ08-S solar pyranometer. The measurements were relocated from 18-m height to 37-m height on February 14, 2017. Global Radiation from the Agia Marina is monitored using a weather station (Campbell Scientific Europe). In Helsinki, the global radiation is measured using Kipp and Zonen CNR1 at 31 m above ground level in the SMEAR III station. In Budapest, global radiation was measured by an SMP3 pyranometer (Kipp and Zonnen, The Netherlands) on the roof of the building complex with a time resolution of 1 min. Its operation was checked by comparing the measured data with those obtained from regular radiation measurements performed by a CMP11 pyranometer (Kipp and Zonnen, The Netherlands) at the Hungarian Meteorological Service (HMS) at a distance of 10 km. The annual mean GlobRad ratio and SD of the 1-h values for the BpART and HMS stations were $1.03\pm0.23$ for GlobRad $> 100$ W m$^{-2}$, which changed to $1.01\pm0.05$ when considering clear sky conditions. In Beijing, GlobRad intensity from 285 nm to 2800 nm was measured at the rooftop of the 5-floor building using a CMP11 pyranometer (Kipp and Zonnen, Delft, The Netherlands). The radiometer was maintained weekly to ensure the location horizontally and clean. In order to do the fitting for the nighttime data, zero values were replaced by the detection limit of the instrument assumed to be half the minimum measured radiation. In Kilpilahti, no global radiation measurements were available, so we relied on radiation data measured at the SMEAR III station which is around 32 km from the measurement site.

## Alkenes

Volatile organic compounds (VOCs) were measured with a proton transfer reaction quadrupole mass spectrometer (PTR-MS, Ionicon Analytik GmbH) in Hyytiälä. Ambient mixing ratios are measured every third hour from several different measurement heights. In this study, we use monoterpene concentration from 16.8 m height. The instrument is calibrated regularly with standard gas (Apel-Riemer Environmental, Inc.) (Taipale et al., 2008). The same instrumentation was used to measure monoterpene concentrations in Kilpilahti every 1 hour.
In Beijing, VOCs were measured using single photon ionization time-of-flight mass spectrometer (SPI-MS 3000R, Hexin Mass Spectrometry) with unit mass resolution (UMR) (Gao et al., 2013) from September 27, 2018 to May 28, 2019. The alkenes included here are butylene, butadiene, isoprene, pentene and hexene. As the instrument cannot distinguish conformers, the pentene and hexene could

also be cyclopentene and cyclohexene. Correlation coefficients between the different variables used in our study in all four locations are shown in Figures S2-S6.

**Meteorological parameters**

Temperature (T) and relative humidity (RH) in Hyytiälä were measured at 16.8 m using a 4-wire PT-100 sensors, and relative humidity sensors (Rotronic Hygromet MP102H with Hygroclip HC2-S3, Rotronic AG, Bassersdorf, Switzerland), respectively. In Agia Marina, T and RH were measured using a weather station (Campbell Scientific Europe). T and RH were measured at the Physicum rooftop 26 m above ground level and 220 m northeast from SMEAR III using a Pentronics PT100 sensor and Vaisala HMP243 transmitter, respectively. In Budapest, T and RH were measured using a Vaisala HMP45D humidity and temperature probe, at the Hungarian Meteorological Service (HMS) within a 10 km radius from the BpArt station. In Beijing, meteorological parameters are monitored by a Vaisala Weather station data acquisition system (AWS310).

**3. Derivation of the new proxy**

We applied the following equation to describe the time-evolution of gas-phase sulphuric acid concentration:

$$\frac{d[H_2SO_4]}{dt} = k_0[OH][SO_2] + k_2[O_3][Alkene][SO_2] - CS[H_2SO_4] - k_3[H_2SO_4]^2 \quad (1)$$

Here, $k_o$ represents the coefficient of $H_2SO_4$ production term due to the well-known $SO_2$ - OH reaction (Petäjä et al., 2009) and $k_2$ is the coefficient of $H_2SO_4$ production via stabilized Criegee Intermediates (sCI) produced by the ozonolysis of alkenes (Mauldin et al., 2012). Here we use available monoterpene concentration (MT) as a proxy for alkenes in Hyytiälä as they are the dominating species in the boreal forest environment (Hakola et al., 2012; Hellén et al., 2018; Rinne et al., 2005). For Beijing, we use urban dominating aromatic alkenes. As no VOC measurements are performed in neither Agia Marina nor Budapest, we evaluate the proxy without the stabilized Criegee Intermediate source term. It is important to note here that the coefficient for sCI is a "bulk" term, and it varies from place to place due to the differences in sCI structures and different production efficiency from different alkene species (Novelli et al., 2017; Sipilä et al., 2014). The third term in Equation 1 represents the loss of $H_2SO_4$ onto pre-existing aerosol particles, known as condensation sink (CS) and is calculated by multiplying the CS calculated for sulphuric acid with the concentration of sulphuric acid monomer. The fourth term in Equation 1 is defined as the square of sulphuric acid concentration multiplied by clustering coefficient $k_3$. The square of sulphuric acid represents the collision of two sulphuric acid monomers forming a sulphuric acid dimer which was found to be the first step of atmospheric cluster formation (Yao et al., 2018). Therefore, this term takes into account the additional loss of $H_2SO_4$ due to cluster formation not included in the term containing CS. This is necessary because CS is only inferred from size-distribution measurements at maximum down to 1.5 nm, i.e. not containing any cluster concentrations and hence losses onto these clusters. This term is written in the form of sulphuric acid dimer production, which seems to be the first step of cluster formation once stabilized by bases (Kulmala et al., 2013; Almeida et al., 2013; Yao et al., 2018).

Since measuring the OH concentration is challenging, we first replaced it with the UVB radiation intensity, which has been shown to be a good proxy for the OH concentration (Berresheim et al., 2002; Lu et al., 2019; Rohrer and Berresheim, 2006). Unfortunately, UVB was not measured in all the field studies considered here. Alternatively, GlobRad, a commonly measured quantity, tends to correlate well with UVB and can generally replace it, as used previously by Petäjä et al. (2009). We confirmed the strong correlation between UVB radiation and Global radiation in two locations, Hyytiälä and Beijing (Figure S7-S8). Accordingly, the coefficient $k_1$ here replaces the coefficient of $H_2SO_4$ production $k_o$ terms (Equation 2). We proceed here using only GlobRad in the proxy to be consistent with the two other locations where UVB was not measured (Agia Marina and Budapest).

$$\frac{d[H_2SO_4]}{dt} = k_1 GlobRad[SO_2] + k_2[O_3][Alkene][SO_2] - CS[H_2SO_4] - k_3[H_2SO_4]^2 \qquad (2)$$

By assuming a steady state between $H_2SO_4$ production and loss, the $H_2SO_4$ concentration can be solved directly from Equation (2):

$$[H_2SO_4] = -\frac{CS}{2k_3} + \left[\left(\frac{CS}{2k_3}\right)^2 + \frac{[SO_2]}{k_3}(k_1 GlobRad + k_2[O_3][Alkene])\right]^{\frac{1}{2}} \qquad (3)$$

In order to evaluate the importance of each of the source terms in determining the change in sulphuric acid concentration, we refitted the data after excluding the stabilized Criegee intermediates source pathway as shown in Equation 4.

$$\frac{d[H_2SO_4]}{dt} = k_1 GlobRad[SO_2] - CS[H_2SO_4] - k_3[H_2SO_4]^2 \qquad (4)$$

In order to evaluate the importance of each of the sink terms in determining the sulphuric acid concentration, we refitted the data after excluding the loss of sulphuric acid via the cluster formation pathway using Equation 5.

$$\frac{d[H_2SO_4]}{dt} = k_1 GlobRad[SO_2] + k_2[O_3][Alkene][SO_2] - CS[H_2SO_4] \qquad (5)$$

we also refitted the data using the simple proxy proposed by Petäjä et al. (2009) by excluding the formation of sulphuric acid via stabilized Criegee intermediates source pathway and loss of sulphuric acid via the cluster formation pathway using Equation 6 and evaluated it by comparing to the original Petäjä et al. (2009) proxy using Equation 7 and Mikkonen et al. (2011) using Equation 8. The calculation of the scaled reaction constant $k$ used in Equation 8 is given in the supplementary material section 1.

$$\frac{d[H_2SO_4]}{dt} = k_1 GlobRad[SO_2] - CS[H_2SO_4] \qquad (6)$$

$$\frac{d[H_2SO_4]}{dt} = 1.4x\ 10^{-7}x\ GlobRad^{-0.7}[SO_2][GlobRad] - CS[H_2SO_4] \qquad (7)$$

$$[H_2SO_4] = 8.21 \; x \; 10^{-3} \; k \; GlobRad[SO_2]^{0.62}(CS.RH)^{-0.13} \qquad\qquad (8)$$

The equations derived for each of the sites can be found in Table 1. The fitting coefficients were obtained by minimizing the sum of the squared logarithm of the ratio between the proxy values and measured sulphuric acid concentration using the method described by Lagarias et al. (1998), a build-in function *fminsearch* of MATLAB, giving the optimal values for the coefficients. The data were subject to 10,000 bootstrap resamples when getting each of the $k$ values as a measure of accuracy in terms of bias, variance, confidence intervals, or prediction error (Efron and Tibshirani, 1994). We accounted for the systematic uncertainty in $H_2SO_4$ and predictor variables. For every bootstrap fit, we assumed both $H_2SO_4$ and all predictor variables to be affected by independent systematic errors between its lower and upper accuracy limits. More details on the bootstrap resampling method and uncertainty introduction can be found in the supplementary information. The 25th percentile and 75th percentiles of the coefficients are shown for all locations together with the median $k$ values in Table 2. The median $k$ values from the bootstrap resamples were used in the equations for deriving sulphuric acid concentrations at each site. Figures S2-S6 present the correlation matrix between the different variables participating in $H_2SO_4$ formation and loss in all locations. In Beijing, the Alkenes (AVOCs) have different patterns in day and night which forces us to have two separate equations for daytime and nighttime. The goodness of the fit and the probability of overfitting or under-fitting was evaluated using the Akaike information criterion (Figure S9), which also compares the proxies given in equations 2, 4, 5 and 6. The criterion uses the sample size (number of points), the number of parameters (terms in the equation) and the sum of squared estimate of errors (SSE: deviations predicted from actual empirical values of data) to estimate the quality of each model, relative to each of the other models and thus provides means for model selection (McElreath, 2018).

## 4. Results and Discussions

### 4.1. The sulphuric acid proxy for Hyytiälä SMEAR II station

Figure 1 shows the scatter plot between the observed $H_2SO_4$ concentrations and that derived by the proxy using the full Equation 2. The correlation coefficient was 0.84 (1860 data points). The data were related to 3-hour medians, as the monoterpene concentration was measured only every third hour. In Figure 1B-D, the proxy is refitted after removing one of the source or sink terms (Equations 4-6), in order to evaluate the sensitivity of the proxy to each of the terms and to show the improvement of the proxy using the additional source and sink (Figure 1A) in comparison to the simple proxy that was used by Petäjä et al. (2009) (Figure 1D). Our results show that the integration of additional terms of $H_2SO_4$ formation (i.e. the stabilized Criegee Intermediates) and loss (atmospheric cluster formation) gives the new proxy the ability to accurately capture the diurnal variation of the $H_2SO_4$ concentration, demonstrating a clear improvement over the earlier physical proxy (Petäjä et al., 2009). In Figure 1B the corresponding data are shown without the alkene term (Equation 4). The correlation is substantially weaker (0.70) than with the full equation. Even more importantly, we cannot estimate the contribution of the alkene term to the sulphuric acid concentration (Figure 2 – Fit 2) as the fit results also in an unphysical coefficient for cluster formation (Kürten et al., 2015) and the fit fails to capture the diurnal pattern during dark hours after 16:00 (Figure 2 – Fit 2). When fitting the data without the cluster source term (Equation 5), the correlation coefficient is high (Figure 1C), yet the goodness of the fit is not as good as when the cluster source term is taken into account (Table S4 - Figure S9). Furthermore, we derived an additional proxy equation using CS corrected for hygroscopic growth (Laakso et al., 2004) to be used when calculating a more robust proxy for Hyytiälä. The details, equation and results are shown in the supplementary information (Figure S10-S12).

Note that we opted for deriving a bulk proxy (daytime and nighttime together) instead of two independent proxies, one for daytime and one for nighttime separately. Our results show that one bulk equation is able to explain the Hyytiälä sulphuric acid daytime and nighttime sources accurately. Additionally, separating the bulk equation into two distinct equations results in bias towards the pattern of one of the predictor variables. For instance, the $k_1$ value during daytime follows the cycle of global radiation, while that of $k_2$ follows the cycle of alkenes. Therefore, in order to accurately reflect the continuum of source and sink terms throughout the day, we decided on the bulk proxy. Additionally, one bulk equation was able to predict sulphuric acid concentrations during daytime and nighttime with high accuracy (slope of ~1) as further discussed in section 4.5.

The fit was able to reproduce the sulphuric acid concentration in such clean environment without the cluster term (Figure 2 – Fit 3), perhaps due to low concentrations of bases participating in clustering in Hyytiälä (Jen et al., 2014). Finally, the corresponding data without both the alkene source term and cluster formation source term (Equation 6, Figure 1D) shows a weaker correlation between the measured and modelled sulphuric acid concentration (0.70), but more importantly, it deviates far from the 1:1 line during both daytime and nighttime (Figure 2 – Fit 4). It is important to note here that when deriving the Petäjä proxy (Petäjä et al. 2009), the model relied on summer data between April and June 2007 which could explain the misfit with the current data from Hyytiälä which spans the whole year. See also figures S13 and S14 for scatter plots comparing the measured sulphuric acid concentrations of the training data set with Petäjä et al. 2009 and Mikkonen et al. 2011, respectively. In general, using all four terms in equation 2 shows improvement over all other combinations (Equations 4-6) in terms of not only correlation coefficients and accurate diurnal cycle between measured and calculated concentrations of sulphuric acid as shown in Figures 1 and 2, but also show

a better goodness of the fit as shown in Table S4 and Figure S9 when using the AIC statistical method.
The final equation for the boreal forest environment can be found in Table 1, Equation 9.

### 4.2. Sulphuric Acid Proxy at a Rural Site: Agia Marina, Cyprus

Since there were no direct measurements of alkenes in Agia Marina, we had to exclude the formation
of $H_2SO_4$ in the oxidation by sCI from the proxy, and therefore we derived only the daytime $H_2SO_4$
proxy concentration. The correlation between the measured and proxy concentration of $H_2SO_4$ was
0.88 (96 data points) which shows that the chosen predictors were able to explain the measured
sulphuric acid concentration largely (Figure 3). However, the slope deviates from the 1-to-1 line
which could be attributed to the additional formation mechanisms that we could not include with the
current data. However, the addition of the cluster loss mechanism shows a noticeable improvement
over the simple proxy, in Figure 3B (R = 0.80). The cluster loss term starts to become more important
in this rural environment in comparison to the boreal forest, which could be due to a higher
concentration of stabilizing bases in Agia Marina compared with Hyytiälä. Although both fits of,
Equation 4 and 6, show similar diurnal patterns (Figure 4, Fits 2 and 4), the loss term due to $H_2SO_4$
cluster formation improved the precision of the new proxy (Figures 3). According to the statistical
AIC method, the goodness of the fit has improved from 70 to 33, with and without the clustering
term, respectively, as shown in Figure S9. Also, even without the alkene term, the newly derived
coefficients improved the proxy in comparison to Petäjä et al. (2009) and Mikkonen et al. (2011) as
shown in Figures 4, S13 and S14. The final equation for the rural site can be found in Table 1,
Equation 10.

### 4.3. Proxy for urban environment: Budapest, Hungary

Next we try to understand the mechanisms of sulphuric acid formation and losses in an even more
complex environment, such as urban Budapest (Figures 5 & 6). Since there were no direct
measurements of alkenes there, neither its proxies such as monoterpenes or anthropogenic volatile
organic compounds, we derived the sulphuric acid proxy excluding the formation due to stabilized
Criegee Intermediate pathway, as in Equation 4. In comparison to the simple proxy (Figure 5B; R =
0.49; 263 data points), the correlation between the measured and proxy concentration of $H_2SO_4$
improved with the addition of the loss term due to cluster formation, R = 0.59 (Figure 5A). The
correlation between measured and modelled values of sulphuric acid became weaker in Budapest in
comparison to Hyytiälä and Agia Marina, which could be attributed to a more complex environment,
and additional pathways of sulphuric acid formation and losses. Additionally, we observed a sudden
$SO_2$ concentration change in the middle of the campaign, possibly due to sudden change in local
meteorology and airmass transport, which could also explain the weaker correlation (See Figure S1).
The loss term due to $H_2SO_4$ dimerization improved the precision of the new proxy in comparison to
the simple model as well as the Petäjä et al. (2009) or the Mikkonen et al. (2011) derivation, as shown
in Figure 6, S13 and S14). We think that the overestimation in the Petäjä proxy is because of its
dependence on the $SO_2/CS$ ratio. The proxy is originally derived in Hyytiälä and when we apply the
same coefficients to Budapest it gives higher estimated concentration compared to the measured since
$SO_2/CS$ ratio is smaller in Budapest (Figure 9). Although the proxy developed by Mikkonen et al.
(2011) has shown to work in varying environments, it clearly overestimates the sulphuric acid
concentration in Budapest for perhaps the same reasons (its dependence on the $SO_2/CS$ ratio). It is

also visible from Figures 5 and 6, that the addition of the dimerization term was capable of better capturing the lower $H_2SO_4$ concentrations in comparison to fitting the data without the dimerization term. In comparison to both Hyytiälä and Agia Marina, the coefficient associated with dimerization in Budapest is slightly higher, which can be attributed to the availability of a possibly facilitated clustering due to higher abundance of stabilizing bases such as amines and ammonia (discussed in section 4.6). The final equation for the urban environment can be found in Table 1, Equation 11.

### 4.4. Proxy for Megacity: Beijing, China

In megacities, in our case Beijing, the sulphuric acid concentration is particularly high during nighttime, which confirms the need for determining the contribution of sources other than OH (radiation) to its formation. Our observations emphasize the contribution of the alkene pathway, as without considering this route we would not replicate morning hours correctly. During daytime, there is enhanced dimerization and cluster formation due to the abundance of stabilizing bases (Yao et al., 2018). We assessed the derivation of the proxy equation first using daytime data and nighttime data separately, and found that such a separation results in an unphysical $k_3$ value since clustering in Beijing happens mostly during daytime(Zhou et al., 2020). This obstacle was also observed when deriving a bulk equation. To overcome it, we set an upper limit for the $k_3$ value at 7 x $10^{-9}$ obtained from the fitting of daytime data (GlobRad >= 50 W/m$^2$). The reason for such an observation is that, in such a complex environment, sulphuric acid might originate from sources other than the ones we accounted for in our calculation especially during nighttime, for example through the hydrolysis of $SO_3$ formed from non-photochemical processes (Yao et al., 2020, In Rev.). The alkenes or volatile organic compounds during daytime are different from those during nighttime, and might vary between seasons, which could be attributed to a different fleet composition during those times or the biogenic activity (Yang et al., 2019). However, the derived equation 12 (derived from spring data) is able to predict the daytime and nighttime sulphuric acid concentrations during summer and autumn (See more in section 4.5)

In Figure 7, we see an improvement of the new proxy (Equation 2) in comparison to the simple proxy (Equation 6) derived by Petäjä et al. (2009) as the former takes into the account the additional sources and sinks of $H_2SO_4$ which were not considered in previous works (See also Figure S9). Introducing the alkene production term improved the accuracy of the $H_2SO_4$ concentration during both daytime and nighttime (Figures 7 and 8), which supports our assumption that $H_2SO_4$ formation during nighttime is driven by stabilized Criegee Intermediates. In Figure 7B we show the proxy without the alkene term is unable to capture the nighttime concentrations. In Figure 9, we see the importance of all sources and sinks predicted for sulphuric acid, as Fit 1 (Equation 2) predicts best the measured sulphuric acid concentration. Additionally, according to the statistical AIC method, using the full equation has the least probability of inaccuracy and error in estimating the sulphuric acid concentration (Figure S9). Moreover, it is clear that the addition of the cluster sink term in Megacity environment is required due to its large contribution as a sink for $H_2SO_4$ especially due to higher concentrations of stabilizing molecules, the cluster mode (sub-3 nm) particle concentration, are the highest in Chinese Megacities (Zhou et al., 2020). The final equation for the megacity can be found in Table 1, Equation 12.

### 4.5. Predictive power of proxy equations

Each of the proxies of the boreal forest environment, rural background and megacity were tested for predictive power on independent data sets using extended data sets from the same location or using measurements from locations with similar characteristics. The sulphuric acid concentrations at each of these locations is modelled using the equation (with median k per source/sink term) relevant to the site and compared to the measured concentrations. The derivation of the sulphuric acid concentrations using 10,000 combinations of k values as well as the error on the predictions are shown in the supplementary information. Note that the testing data sets are not subject to any boot strap resampling or uncertainty additions, but are rather used as is for testing the predictive power of the suggested proxy.

4.5.1 Boreal forest environment: Hyytiälä

For testing the predictive power of the boreal forest proxy (Equation 9), we use an independent testing data set from the same location measured from January 1, 2017 to June 5, 2017. Results show that the modelled sulphuric acid concentrations correlate well (R = 0.7) with the measured sulphuric concentrations with a slope of 0.997 for the testing data set (Figure 10A and S16). Moreover, we tested the four fits on the testing data set; i.e. the full Equation 2, the equation without the Stabilized Criegee Intermediates source (Equation 4), the equation without the cluster sink term (Equation 5) and the equation without neither the Stabilized Criegee Intermediates source nor the cluster sink term (Equation 6), and found that Fit 1 (Equation 4) best defines the measured sulphuric acid concentration in comparison to the rest of the equations (Figure S17). The diurnal cycle is also accurately described by the Equation 4 which captures both nighttime and daytime (Figure S18).

4.5.2. Semi-urban location: Helsinki

For testing the predictive power of the rural background site proxy (Equation 10), we use an independent testing data set from a semi-urban location in Helsinki, Finland measured from July 1, 2019 to July 16, 2019 during daytime (GlobRad >= 50 W/m$^2$). The rural background site equation 10 is used as the condensation sink and $SO_2$ concentrations in the testing location are within the interquartile span of the Agia Marina measurements (Figure 9, Table S3). Results show that although the modelled sulphuric acid concentrations do not correlate as well as in other locations (R = 0.44), the bias could be attributed to the missing source (alkene) in the original equation (Figure 10B). Indeed, looking at the binned data, we find that at within each concentration bin the modelled sulphuric concentrations tend to span the 1:1 line. Actually, the discrepancy between the measured and the modelled concentration is smaller than the model prediction error (Figure S19). Note that the model prediction error is estimated as the interquartile range of the modelled $H_2SO_4$ concentration of a single point in time arising from the uncertainty in k values. For the rural background site, we also found that the diurnal cycle is better described when introducing the additional clustering sink term (Figure S20).

4.5.3. Megacity: Beijing

For testing the predictive power of the megacity proxy (Equation 12), we use an independent testing data set from the same location (Beijing) measured from September 1, 2019 to October 15, 2019. Results show that the modelled sulphuric acid concentrations correlate well (R = 0.83) with the measured sulphuric concentrations with a slope of ~1.1 for the testing data set (Figure 10C). Also for this site, we tested the four fits on the testing data set; i.e. the full Equation 2, the equation without the Stabilized Criegee Intermediates source (Equation 4), the equation without the cluster sink term (Equation 5) and the equation without neither the Stabilized Criegee Intermediates source nor the cluster sink term (Equation 6), and found that Fit 1 (Equation 4) best defines the measured sulphuric acid concentration in comparison to the rest of the equations (Figure S22). The diurnal cycle is also described by the Equation 4 which captures both nighttime and daytime (Figure S23).

4.5.4. Industrial area: Kilpilahti

Finally, we tested the predictive power of our developed proxy on a data set measured at an industrial
area in close proximity to an oil refinery. Interestingly, the median CS at the location lies within the
interquartile range of the CS measured in Hyytiälä and that measured in Agia Marina (Table S3,
Figure 9). The $SO_2$ concentrations at the measurement site are higher than in both Hyytiälä and Agia
Marina, but smaller than the ones reported in Budapest. Additionally, we observed alkene
concentrations at Kilpilahti, which are within the range of those monitored in Hyytiälä attributed to
the green belt in the area (Sarnela et al., 2015). Accordingly, we test the proxy equation 9 on the
Kilpilahti data set. Our results show that Equation 9 is able to predict the sulphuric acid concentrations
in Kilpilahti with a high correlation coefficient (R= 0.74) (Figure 10D). Similar to other locations,
the Fit 1 (Equation 4) best describes the sources and sinks at the location (Figure S25). The
discrepancy between the measured and the modelled concentration is smaller than the model
prediction error for less than 50% of the data points only (Figure S24). This observation is consistent
with the diurnal cycle (Figure S26). During certain mornings (4:00 – 8:00 LT), when the measured
sulphuric concentrations are particularly high, the model was unable to predict the concentrations
accurately. These high concentrations were attributed to air masses coming from the oil refinery
(Sarnela et al., 2015). Indeed, our proxy was not able to explain these morning peaks using biogenic
alkenes, however, in such an industrial area, anthropogenic sources could play a role in determining
the magnitude of sulphuric acid concentrations. With the condensation sink being rather low (median
~0.005 $s^{-1}$), the impact of direct $H_2SO_4$ emissions cannot be ruled out either.

4.6. **Sensitivity of the proxy to the $H_2SO_4$ sources and sinks**

The variations of coefficients related to Equation 3 can be used to get insights into the general
chemical behavior under current atmospheric conditions, as well as into the mechanisms of sulphuric
acid formation and losses in various environments. The contribution of different terms in different
locations seem to vary significantly. The new loss term taking into account clustering starting from
dimer formation needs to be taken into account in all the environments in daytime. On the other hand,
without alkene term it is in practice impossible to get nighttime concentrations correct.

In Table 2, we have presented the fitted coefficients (Equation 3) for all our sites, whereas the
contributions of the different terms in the balance equation are given during daytime in Figure 11 and
Table 3. The contribution of the various source and sink terms to the change of $H_2SO_4$ concentrations
are determined using Equation 2. The median derived $k_1$, $k_2$ and $k_3$ values, together with the measured
$H_2SO_4$, CS, trace gases and GlobRad per site, were used to calculate each of the terms. Source term
1 refers to $k_1$ x GlobRad x $[SO_2]$, source term 2 refers to $k_2$ x $[O_3]$ x [Alkene] x $[SO_2]$, sink term 3
refers to $k_3$ x $[H_2SO_4]^2$ and sink term 4 refers to CS x $[H_2SO_4]$. The contribution of each term is then
calculated as the median or percentiles of the normalized term to the sum of all terms. The variability
of the coefficients (Table 2), as well as the relative contributions of each term to the total sulphuric
acid concentration (Table 3), could give valuable information on the mechanisms resulting in
sulphuric acid formation and losses. At steady state (Equation 2), the sources and sinks are in balance
with each other during both daytime and nighttime, but there were clear differences in the individual
contributions. For instance, a variation in $k_1$ could be due to variations in OH sources and sinks.
Although in urban locations OH sinks are expected to be higher and therefore $k_1$ to be lower,
additional sources of OH are available in such locations, for example HONO (Zhang et al., 2019).
The alkene/Criegee intermediate term was found to be an important $H_2SO_4$ source (Figures 1, 2, 7
and 8), as without it we are not able predict night or morning concentrations of $H_2SO_4$ properly. The
alkene source term contributed up to almost 100% of the $H_2SO_4$ sources during nighttime in Beijing
and up to 90% of the sources during nighttime in Hyytiälä (Figure 12). The Criegee intermediate term
showed its importance mostly when global radiation is low, not only in nighttime but also during
winter (Figure 12) in both Hyytiälä and Beijing. It is important to note here that Criegee intermediates
vary between locations, they also form in different yield percentages from different alkenes (Novelli
et al., 2017; Sipilä et al., 2014). These stabilized Criegee intermediates also react differently under
different environmental conditions.
The CS term had the highest contribution to the total sink in Hyytiälä. Its contribution decreased when
moving towards more polluted environments (Figure 11), to become in Beijing, regardless of the
relatively high condensation sink in Megacities, smaller than that of the cluster sink term (Laakso et
al., 2006; Monkkonen et al., 2005; Monkkonen et al., 2004; Yao et al., 2018).. This observation might
be attributed to decreased effectiveness of condensation sink in more polluted environments (Kulmala
et al., 2017), but also to increased contribution of the clustering sink term in such environments where
the concentration of stabilizing bases is highest, particularly in daytime (Yao et al., 2018; Yan et al.,
2018). It should be noted that measurements of ammonia and similar bases are rare, so their exact
contribution is difficult to estimate. The cluster term is found to contribute most during spring daytime
in Hyytiälä (Figure 12 – A & C), which is the time window during which clustering and thus new
particle formation events happen (Dada et al., 2018; Dada et al., 2017). The same is observed for
Beijing, where the clustering term contributed up to 70% of the total sink terms during daytime
(Figure 12-D) especially during summer when the CS is lowest (Deng et al., 2020).

**5. Conclusions and recommendations**

Sulphuric acid is a key gas-phase compound linked to secondary aerosol production in the
atmosphere. The concentration of sulphuric acid in the gas phase is governed by source and sink
terms. In this paper we define the sources and sinks of $H_2SO_4$ and derived a physically and chemically
sound proxy for the sulphuric acid concentration using measurements at 4 different locations,
including boreal forest environment (Hyytiälä, Finland), a rural Mediterranean site (Cyprus), an urban
area (Budapest) and a megacity (Beijing). When describing the change in gas phase sulphuric acid
concentration, we took into account two source terms: 1) photochemical oxidation of sulfur dioxide
and 2) sulphuric acid originating from alkene and ozone reactions and associated stabilized Criegee
radical pathway. For the sink terms, we considered 3) the loss rate to the pre-existing aerosol described
by condensation sink, and 4) loss rate of sulphuric acid monomer due to clustering process.
In general, the variation in the environmental conditions and difference in concentrations of air
pollutants affects the coefficients derived and therefore it is important to derive location specific
coefficients. The derived coefficients give insights into the general chemical behavior and into the
mechanisms of sulphuric acid formation and losses in various environments. As improvements from
previously derived proxies, without the alkene $H_2SO_4$ formation pathway, it is in practice impossible
to get nighttime concentrations. On the other hand, the additional loss term taking into account
clustering starting from dimer formation needs to be taken into account in all the environments
especially those with higher cluster formation probabilities due to availability of stabilizing bases.
The coefficients derived do not differ substantially between the different locations. The proxy could
therefore be used at locations with no prior $H_2SO_4$ measurements, provided that the environmental
conditions are approximately similar to those in one of the four sites described here. More specifically,
the proxies could be utilized to derive long-term data sets for $H_2SO_4$ concentrations, which would be

essential in performing various kinds of trend analyses. In order to derive the long term sulphuric acid concentrations, we recommend deriving in-house coefficients in case sulphuric acid concentrations are directly measured rather than using the ones from already derived studies. The choice of equation depends on the availability of the data on site. In case alkenes or their proxies are measured and sulphuric acid is measured, derivation of the coefficients should be based on Equation 2. In case neither alkenes nor their proxies are measured but sulphuric acid is measured, the coefficients and therefore the proxy for daytime only can be derived, using Equation 4. In case, sulphuric acid is not measured, one can calculate the sulphuric acid proxy using the Equation 2 or Equation 4, depending on whether the alkene data is available or not, respectively, using the coefficients suggested in Table 1 which are relevant to the site of interest. In order to make the best choice for the coefficients, Figure 9 can be followed in order to decide which description fits the location of interest best. For instance, in case the condensation sink is between $2 \times 10^{-3}$ and $6 \times 10^{-3}$ s$^{-1}$, and the SO$_2$ concentration is lower than $2 \times 10^9$ molecules. cm$^{-3}$, coefficients of Hyytiälä or the boreal forest are to be used.

**Data availability**

The data used in the manuscript and the MATLAB code which provides the k values are available from the first author at lubna.dada@helsinki.fi.

**Author contributions**

MK came up with the idea, LD, IY, CL, RB analyzed the data, YG, CD, RY, CY, LY, JJ, YL, BC, ZL, YW performed the measurements in Beijing and pre-processed the raw data, NS, TJ, MS, TP performed the measurements in Hyytiälä and pre-processed the raw data, LD, TN, JK, KRD, DS, TH, PP, FB, VMK, MK provided useful discussion and ideas, IS, TW, RB, TJ performed the measurements in Budapest and pre-processed the raw data, MP, JS, RB, TJ performed the measurement in Agia Marina and pre-processed the raw data. RCT, TJ, MS performed the sulphuric acid measurements in Helsinki and pre-processed the raw-data. LD and KRD introduced the error and bootstrap resampling analyses. LD, VMK and MK wrote the manuscript. All co-authors contributed to reviewing the manuscript and to the discussions related to it.

**Competing interests**

All authors declare no competing interests.

**Acknowledgements**

This project has received funding from the ERC advanced grant No. 742206, ERC-StG No. 714621, the Academy of Finland Center of Excellence project No. 307331, Academy of Finland project No. 316114 and 296628, the National Natural Science Foundation of China project No. 41877306 and from National Key R&D Program of China (2017YFC0209503). This project receives funding from the European Union's Horizon 2020 research and innovation program under grant agreements (ACTRIS) No. 654109 and 739530. Funding by the National Research, Development and Innovation Office, Hungary (K116788 and K132254) is acknowledged. We thank V. Varga and Z. Németh of the Eötvös University for their help in the experimental work in Budapest, K. Neitola and T. Laurila for their help at Agia Marina, LJJ. Quéléver and T. Lehmusjärvi for their help in setting up the sulphuric acid measurement in Helsinki. This publication has been produced within the framework of the EMME-CARE project which has received funding from the European Union's Horizon 2020

Research and Innovation Programme, under Grant Agreement No. 856612 and the Cyprus

Government. The sole responsibility of this publication lies with the author. The European Union is
not responsible for any use that may be made of the information contained therein.


 **Tables and Figures**



*Table 1 Equations for sulphuric acid proxy derivation at each of the measurement locations.*

$$[H_2SO_4]_{boreal} = -\frac{CS}{2 \, x \, (4.2 \, x \, 10^{-9})}$$
$$+ \left[ \left( \frac{CS}{2 \, x \, (4.2 \, x \, 10^{-9})} \right)^2 + \frac{[SO_2]}{(4.2 \, x \, 10^{-9})} (8.6 \, x \, 10^{-9} \, x \, GlobRad + 6.1 \, x \, 10^{-29} [O_3][Alkene]) \right]^{1/2} \quad (9)$$

$$[H_2SO_4]_{rural} = -\frac{CS}{2 \, x \, (2.2 \, x \, 10^{-9})} + \left[ \left( \frac{CS}{2 \, x \, (2.2 \, x \, 10^{-9})} \right)^2 + \frac{[SO_2]}{(2.2 \, x \, 10^{-9})} (9.7 \, x \, 10^{-8} \, x \, GlobRad) \right]^{\frac{1}{2}} \quad (10)$$

$$[H_2SO_4]_{urban} = -\frac{CS}{2 \, x \, (9.8 \, x \, 10^{-9})} + \left[ \left( \frac{CS}{2 \, x \, (9.8 \, x \, 10^{-9})} \right)^2 + \frac{[SO_2]}{(9.8 \, x \, 10^{-9})} (1.57 \, x \, 10^{-9} \, x \, GlobRad) \right]^{\frac{1}{2}} \quad (11)$$

$$[H_2SO_4]_{megacity} = -\frac{CS}{2 \, x \, (7.0 \, x \, 10^{-9})}$$
$$+ \left[ \left( \frac{CS}{2 \, x \, (7.0 \, x \, 10^{-9})} \right)^2 + \frac{[SO_2]}{(7.0 \, x \, 10^{-9})} (1.94 \, x \, 10^{-8} \, x \, GlobRad + 1.44 \, x \, 10^{-29} [O_3][Alkene]) \right]^{1/2} \quad (12)$$



Table 2: Coefficients used in the proxy equation in all four environments. Numbers in parenthesis
represent the 25[th] and 75[th] percentiles of boot strapped data, respectively. See supplementary section
2 for more details.

| Location | $GlobRad$ (W/m$^2$) | $k_1$($10^{-8}$ m$^2$ W$^{-1}$ s$^{-1}$) | $k_2$($\cdot 10^{-29}$ cm$^6$ s$^{-1}$) | $k_3$ ($\cdot 10^{-9}$ cm$^3$ s$^{-1}$) |
|---|---|---|---|---|
| Hyytiälä | >0 | 0.85(0.60-1.21) | 6.10(4.27-8.57) | 4.26(2.98-5.99) |
| Agia Marina | >= 50 | 0.92(0.64-1.34) | N/A | 2.21(1.27-3.79) |
| Budapest | >= 50 | 0.16(0.09-0.27) | N/A | 9.80(9.79-9.81) |
| Beijing | > 0 | 1.94(1.12 – 3.50) | 1.45(0.93 – 2.26) | 7.0 |



Table 3: Fraction of each source and sink term to the change in H$_2$SO$_4$ concentration. Median of boot
strap resampling results and their 25[th] and 75[th] percentiles are shown.

| | GlobRad (W/m$^2$) | Source Terms | | Sink Terms | |
|---|---|---|---|---|---|
| | | $k_1$Glob[SO$_2$] | $k_2$[O$_3$][A][SO$_2$] | $-k_3$[H$_2$SO$_4$]$^2$ | $-CS$[H$_2$SO$_4$] |
| Hyytiälä | >0 | 0.34 (0.10-0.44) | 0.16 (0.08-0.40) | 0.16 (0.08-0.26) | 0.34 (0.24-0.42) |
| Agia Marina | >= 50 | 0.5 | 0 | 0.24 (0.19-0.29) | 0.26 (0.21-0.31) |
| Budapest | >= 50 | 0.5 | 0 | 0.26 (0.18-0.31) | 0.24 (0.19-0.32) |
| Beijing | > 0 | 0.28 (2E-4– 0.41) | 0.22 (0.09 – 0.50) | 0.29 (0.19 – 0.39) | 0.21 (0.11 – 0.31) |




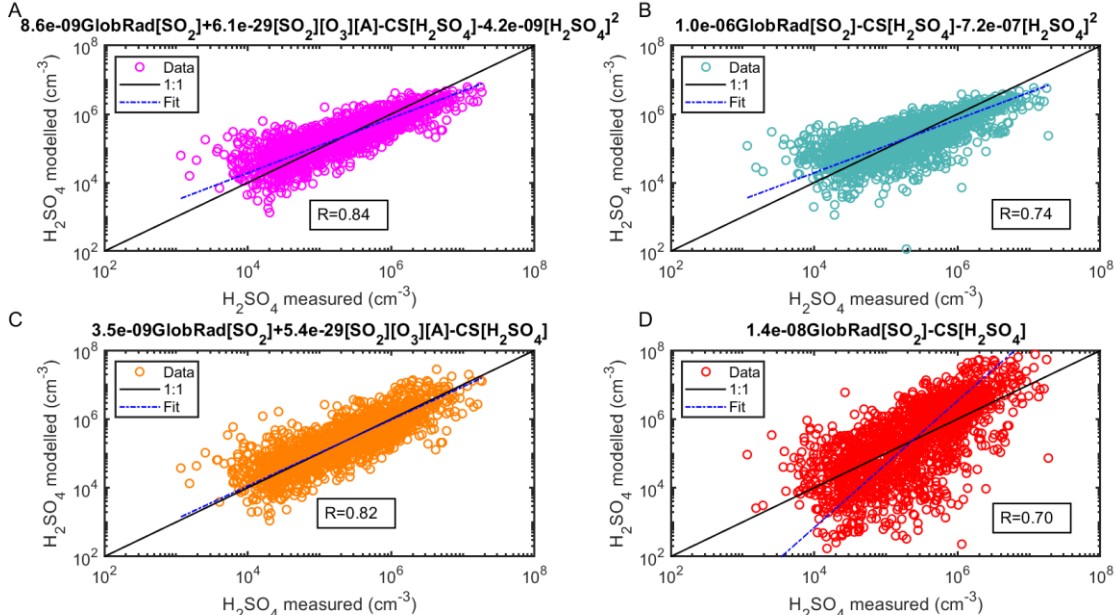

Figure 1: Sulphuric acid proxy concentration as a function of measured sulphuric acid. Observation at SMEAR II station, Hyytiälä Finland. The observed concentrations from the training data set are measured 2016-2019 using CI-APi-ToF and are 3-hour medians resulting in a total of 1860 data points. In (A), the full Equation 2 is used, in (B) the equation without the Stabilized Criegee Intermediates source (Equation 4), in (C) the equation without the cluster sink term (Equation 5) and in (D) the equation without both the Stabilized Criegee Intermediates source and the cluster sink term (Equation 6). The 'Fit' refers to the fitting between the measured and the proxy calculated sulphuric acid concentration ($log(y) = a.log(x)+b$).

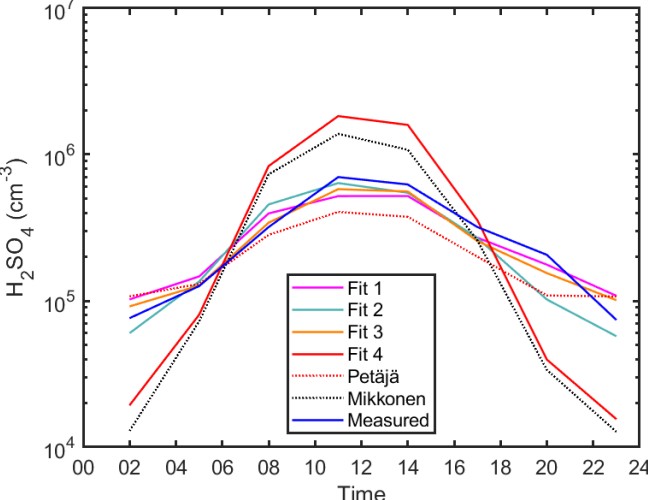

Figure 2: The diurnal variation of sulphuric acid proxy concentrations using different fits and observed concentrations at SMEAR II in Hyytiälä, Finland. Median values are shown. Fits 1,2, 3 and 4 corresponds to the Equations 2, 4, 5, and 6, respectively. Petäjä fit shown is applied using the coefficients reported in Petäjä et al. 2009 (Equation 7). Mikkonen fit shown is applied using the coefficients reported in Mikkonen et al. 2011 (Equation 8).

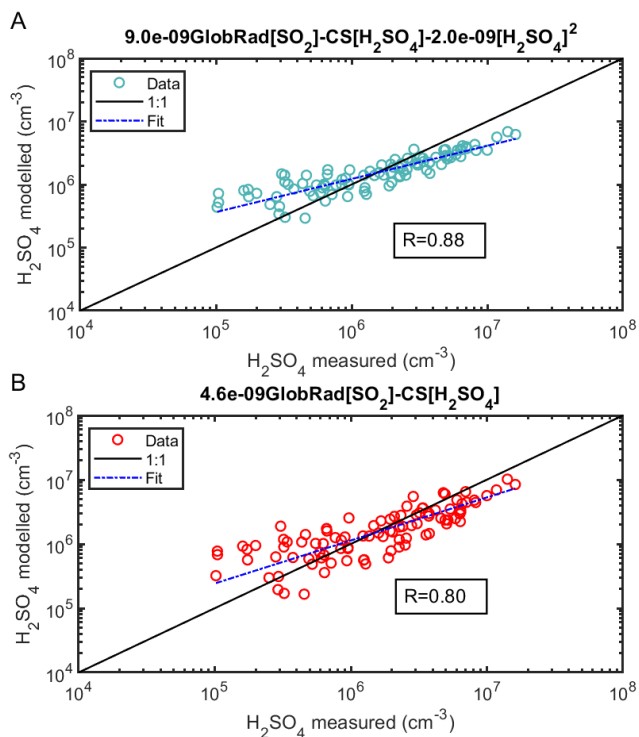

A

$9.0\text{e-}09\text{GlobRad}[SO_2]\text{-}CS[H_2SO_4]\text{-}2.0\text{e-}09[H_2SO_4]^2$

B

$4.6\text{e-}09\text{GlobRad}[SO_2]\text{-}CS[H_2SO_4]$

*Figure 3: Sulphuric acid proxy concentration as a function of measured sulphuric acid. Observation at Agia Marina, Cyprus, excluding the Alkene term. The observed numbers concentrations are measured during Feb- Mar 2018 using CI-APi-ToF and are hourly medians resulting in a total of 96 data points. Sulphuric acid proxy concentration as a function of measured sulphuric acid. In (A), the equation without the Stabilized Criegee Intermediates source (Equation 4) and in (B) the equation without both the Stabilized Criegee Intermediates source and the cluster sink term (Equation 6). The 'Fit' refers to the fitting between the measured and the proxy calculated sulphuric acid concentration ($log(y) = a.log(x)+b$).*

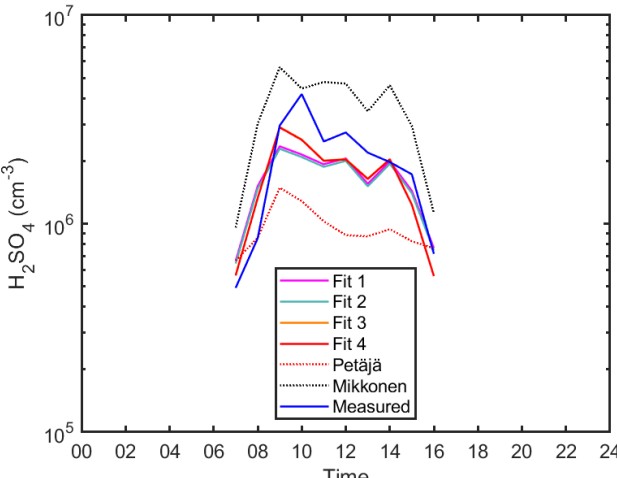

*Figure 4 The diurnal variation of sulphuric acid proxies and observed concentrations in Agia Marina, Cyprus. Hourly median values are shown. Fits 2 and 4 corresponds to the Equations 4 and 6, respectively, See also Figure 3A and B, respectively. Petäjä fit shown is applied using the coefficients reported in Petäjä et al. 2009 (Equation 7). Mikkonen fit shown is applied using the coefficients reported in Mikkonen et al. 2011 (Equation 8).*

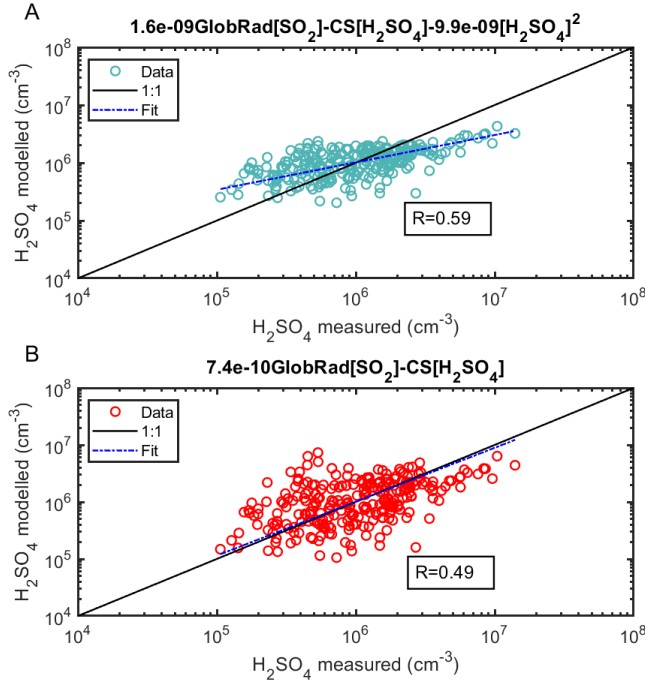

*Figure 5 Sulphuric acid proxy as a function measured sulphuric acid at Budapest station, excluding*
*the Alkene term. The observed numbers are measured during spring 2018 using CI-APi-ToF and are*
*1-hour medians coinciding with the measurement of trace gases and Global radiation every one hour*
*resulting in a total of 263 data points. In (A), the equation without the Stabilized Criegee*
*Intermediates source (Equation 4) and in (B) the equation without both the Stabilized Criegee*
*Intermediates source and the cluster sink term (Equation 6). The 'Fit' refers to the fitting between*
*the measured and the proxy calculated sulphuric acid concentration (log(y) = a.log(x)+b).*

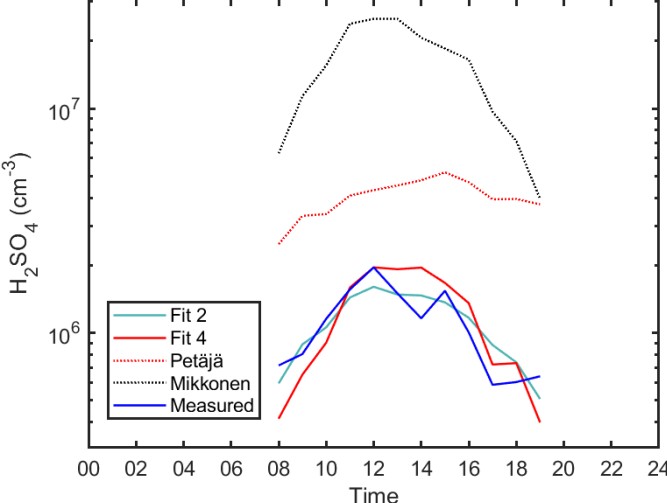

*Figure 6 The diurnal variation of sulphuric acid proxies and measured concentrations in Budapest.*
*Hourly median values are shown. Fits 2 and 4 corresponds to the Equations 4 and 6, respectively.*
*Petäjä fit shown is applied using the coefficients reported in Petäjä et al. 2009 (Equation 7).*
*Mikkonen fit shown is applied using the coefficients reported in Mikkonen et al. 2011 (Equation 8).*

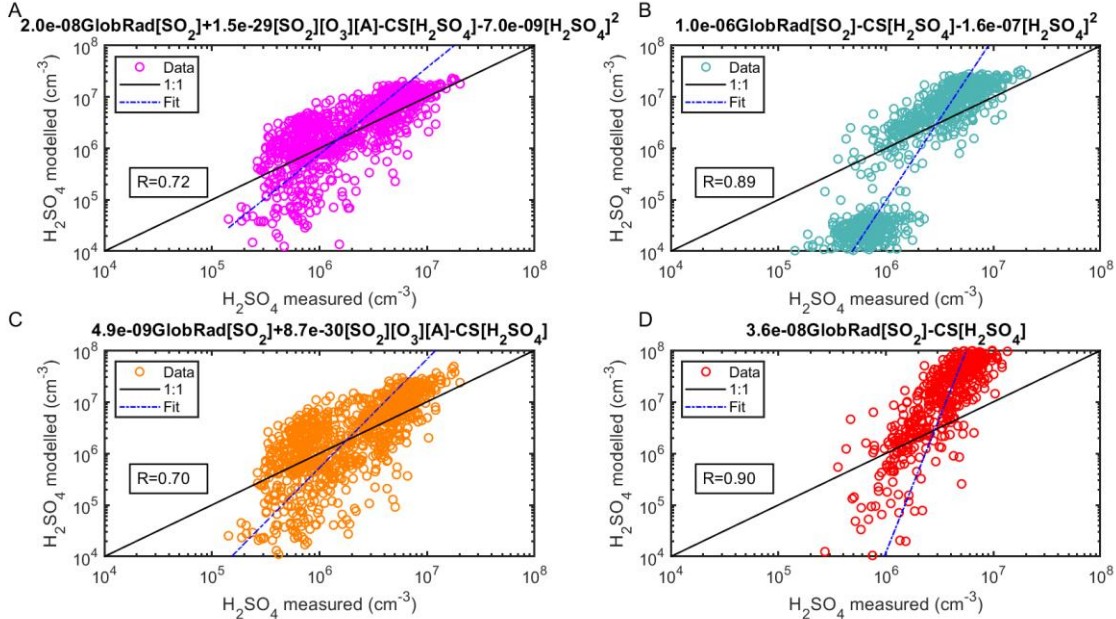

*Figure 7 (A) Sulphuric acid proxy concentration as a function of measured sulphuric acid.*
*Observation at Beijing, China. The observed concentrations of the training data set are measured in*
*2019 using CI-APi-ToF and are 1-hour medians resulting in a total of 877 data points. In (A), the*
*full Equation 2 is used, in (B) the equation without the Stabilized Criegee Intermediates source*
*(Equation 4), in (C) the equation without the cluster sink term (Equation 5) and in (D) the equation*
*without both the Stabilized Criegee Intermediates source and the cluster sink term (Equation 6).*
*Coefficients shown on top of the subplots relate to the daytime values. The 'Fit' refers to the fitting*
*between the measured and the proxy calculated sulphuric acid concentration (log(y) = a.log(x)+b).*

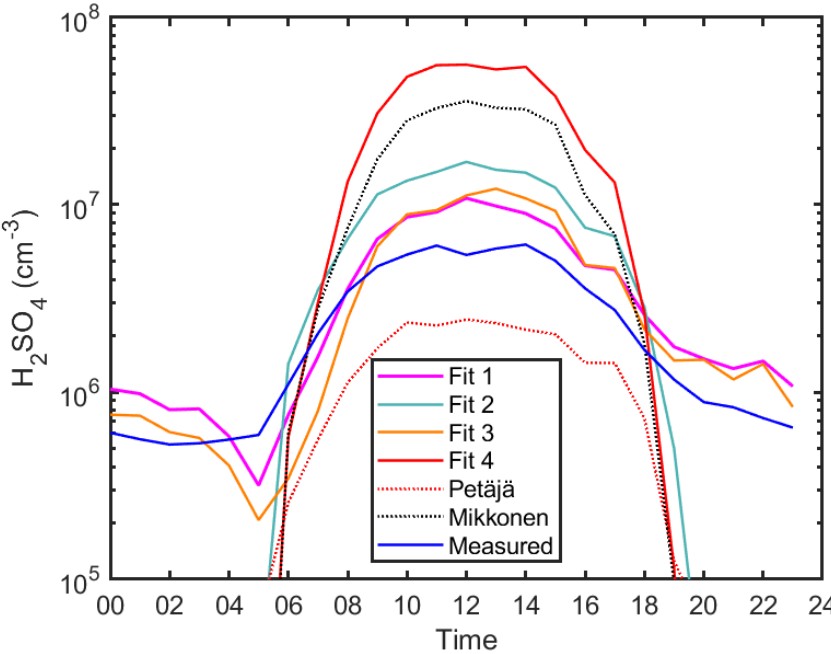

*Figure 8 The diurnal variation of sulphuric acid proxy concentrations using different fits and*
*observed concentrations at Beijing China, Finland. Median values are shown. Fits 1,2, 3 and 4*
*corresponds to the Equations 2, 4, 5, and 6, respectively. Petäjä fit shown is applied using the*
*coefficients reported in Petäjä et al. 2009 (Equation 7). Mikkonen fit shown is applied using the*
*coefficients reported in Mikkonen et al. 2011 (Equation 8).*

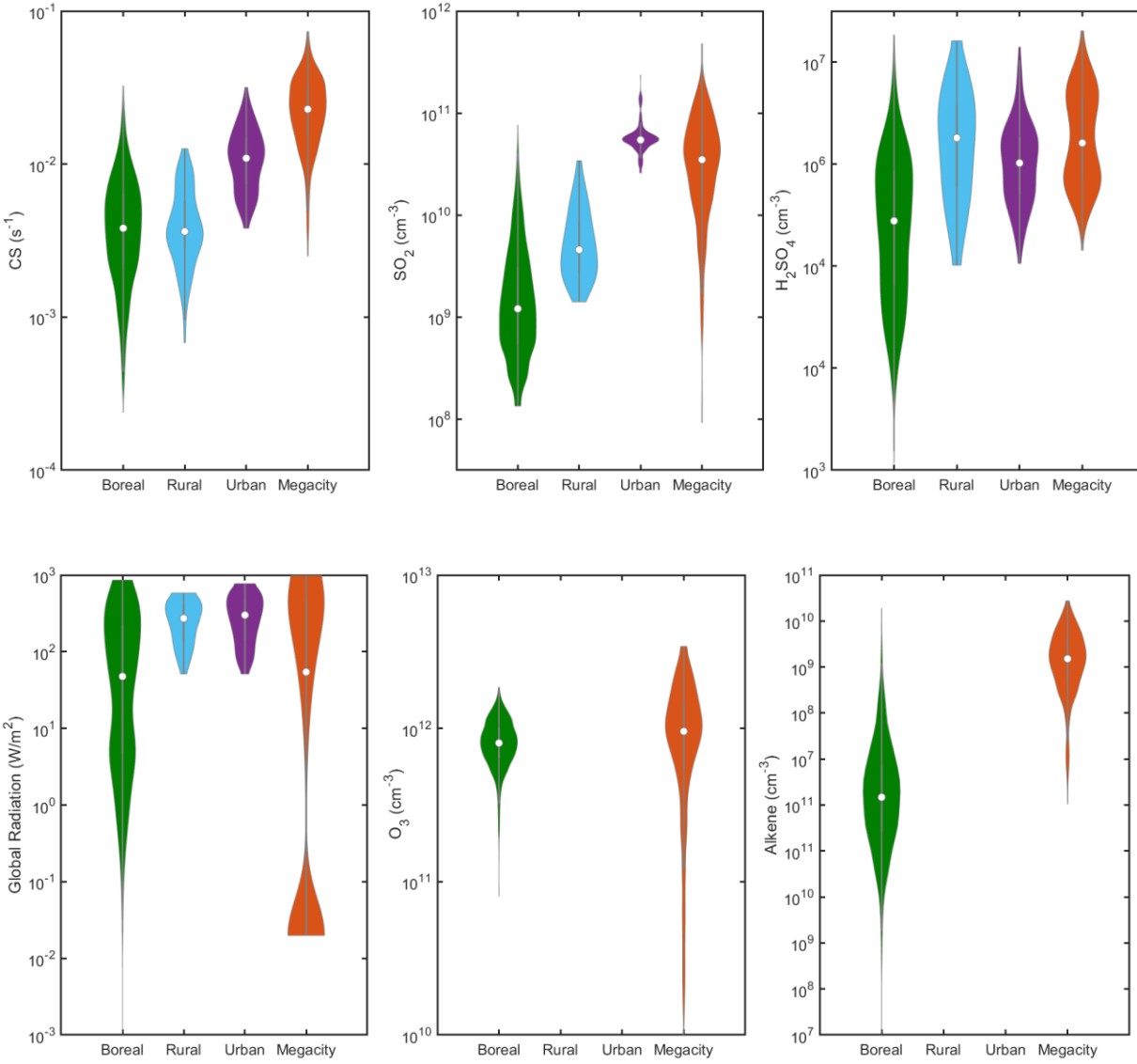

Figure 9 Characteristic predictor variables and $H_2SO_4$ concentrations in diffrerent environement.s $O_3$ and Alkenes data are available from the boreal forest (Hyytiälä) and megacity (Beijing) environments. This figure could be used in order to choose the equation and coefficients for calculating sulphuric acid proxy at a new location. The alkenes in the boreal environment are monoterpenes(e.g. alpha-pinene) and in the Megacity are anthropogenic volatile organic compounds (butylene, butadiene, isoprene, pentene and hexene). The concentrations are displayed as violin plots which are a combination of boxplot and a kernel distribution function on each side of the boxplots. The white circles define the median of the distribution and the edges on the inner grey boxes refer to the $25^{th}$ and $75^{th}$ percentiles respectively. Whole day data is shown for Hyytiala and Beijing, while daytime data (GlobRad > 50 $W/m^2$) for Agia Marina and Budapest. Daytime data (GlobRad > 50 $W/m^2$) is shown in Figure S15. The correlations between the different variables at each site are shown in Figures S2 – S6.

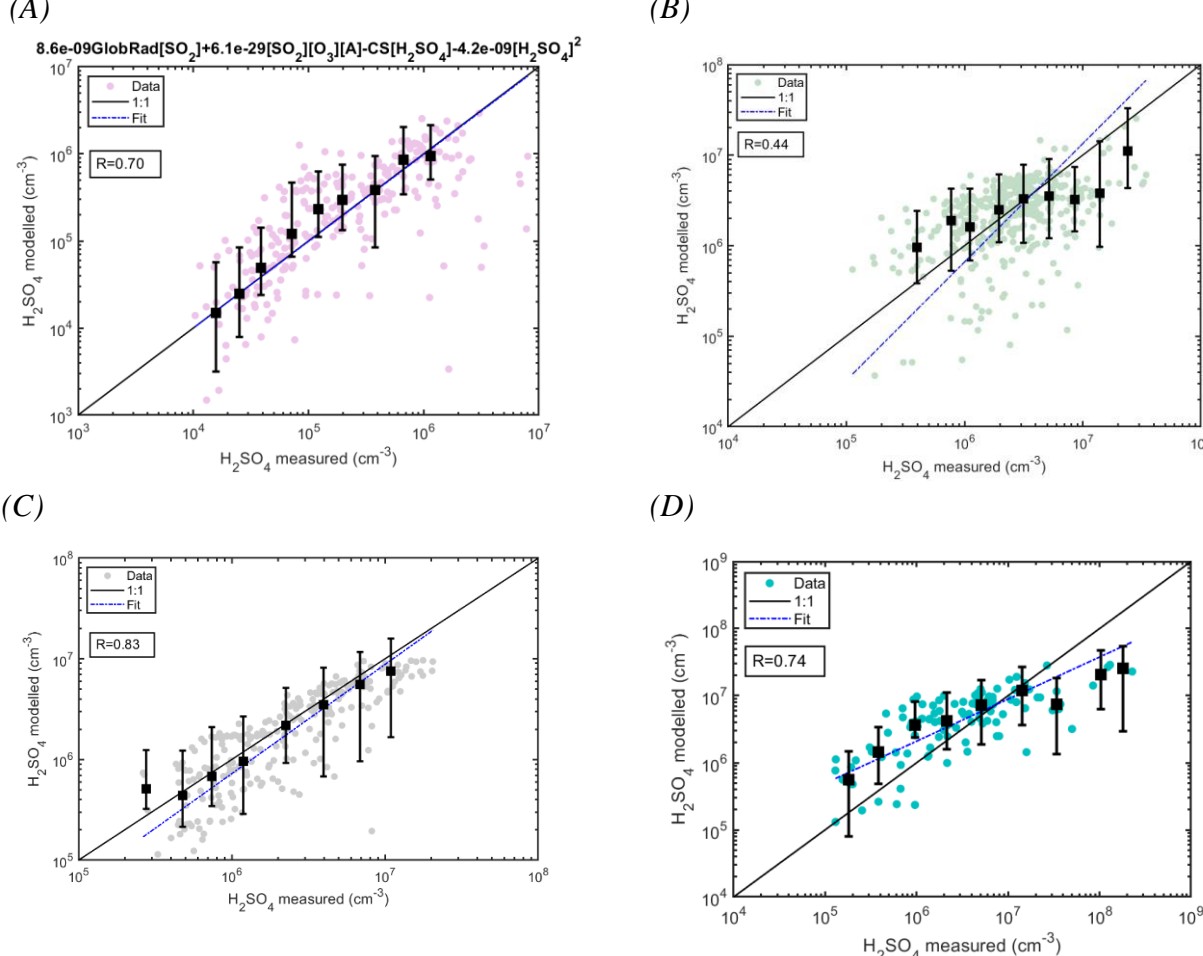

*Figure 10 Sulphuric acid concentrations modelled as a function of measured sulphuric acid using testing data sets. The colored data points refer to the modelled (predicted) concentrations, the dashed blue line refers to the fit ($log(y) = a.log(x)+b$) of the aforementioned data points. The black squares are the median modelled concentrations in logarithmically-spaced measured sulphuric acid bins and their lower and upper whiskers correspond to 25th and 75th percentiles of the predicted concentrations. (A) Hyytiälä SMEAR II station: the concentrations shown are 3-hour medians coinciding with the alkene measurements every three hours resulting in a total of 257 data points. The modelled concentrations are derived using equation 9. (B) Helsinki SMEAR III station: the concentrations shown are 1-hour medians resulting in a total of 416 data points. The modelled concentrations are derived using equation 10. (C) Beijing: the concentrations shown are 1-hour medians resulting in a total of 268 data points. The modelled concentrations are derived using equation 12. (D) Kilpilahti: the concentrations shown are 1-hour medians resulting in 114 data points. The modelled concentrations are derived using equation 9.*

(A)                                              (B)

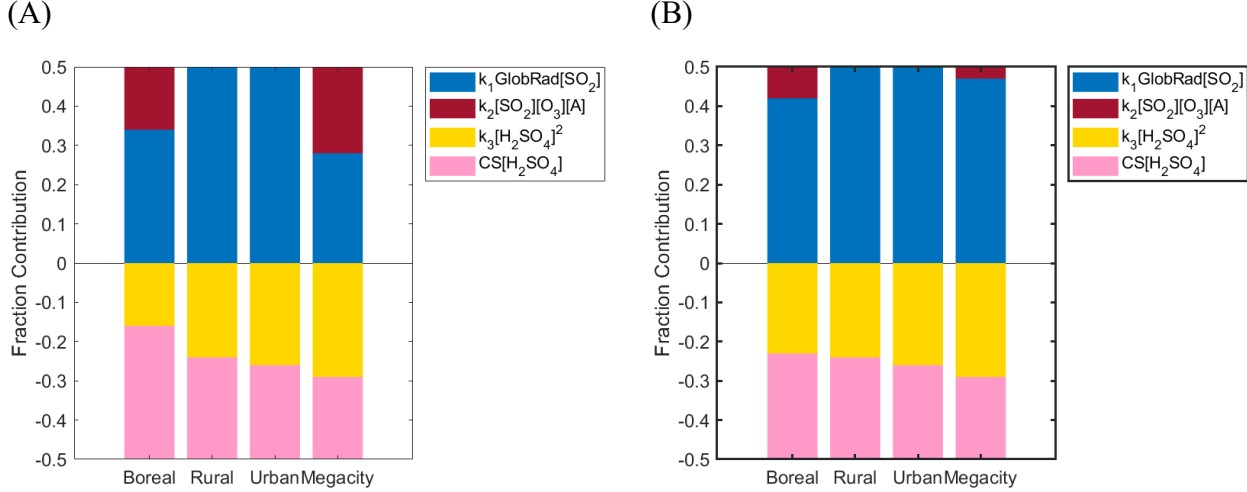

800

*Figure 11 Fraction contribution of each source and sink term to the change in $H_2SO_4$ concentration.*
*Figure 11 is complementary to Table 3. The boreal, rural, urban and megacity labels refer to*
*Hyytiälä, Agia Marina, Budapest and Beijing sites, respectively. Note that the fraction of the alkene*
*term contribution is not zero for the rural or urban sites, but is due to unavailable alkene data from*
*these sites. In (A) we show all day medians for Hyytiälä and Beijing and in (B) we show daytime*
*medians for all sites.*

807

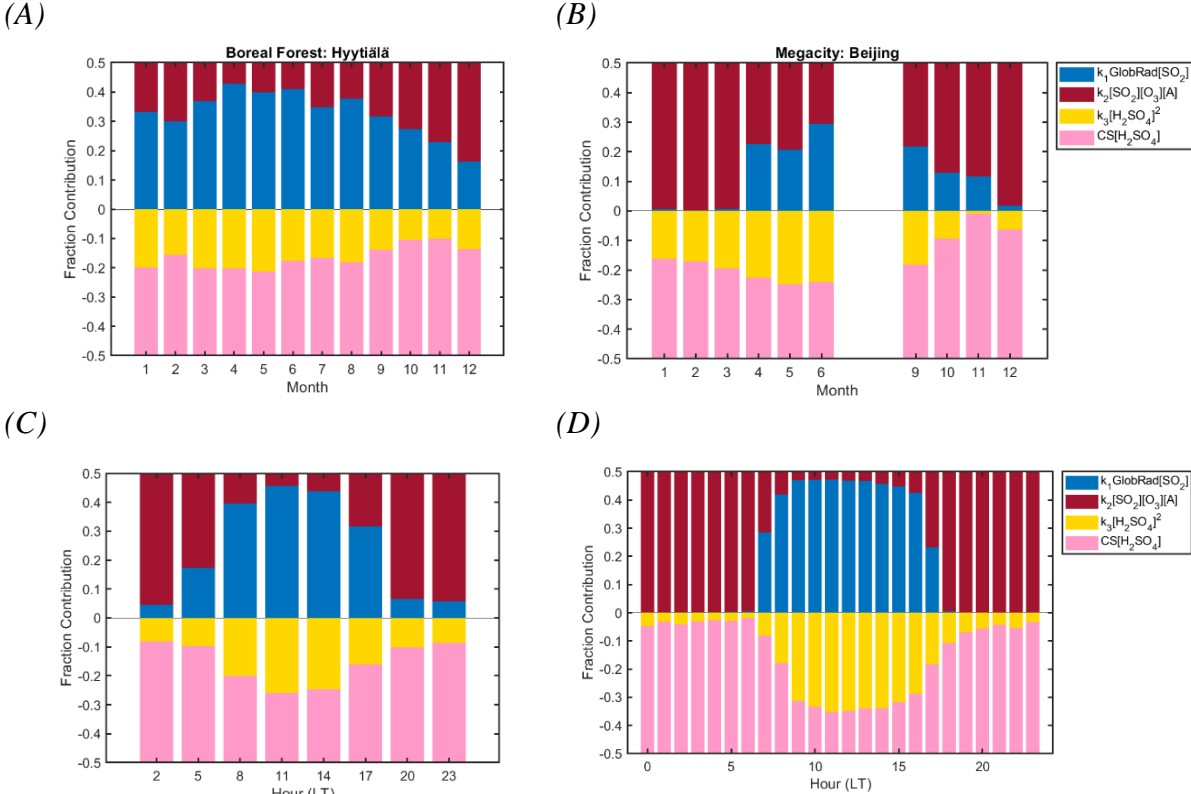

Figure 12 (A) Monthly variation of each source and sink term fraction contribution to the change in $H_2SO_4$ concentration in Hyytiälä within the training data set 2016-2019. (B) Monthly variation of each source and sink term to the change in $H_2SO_4$ concentration in Beijing within the training and testing data sets 2019, the data outside the training and testing data sets has missing measured sulphuric acid concentrations, so proxy concentrations were used in obtaining this figure. (C) Diurnal variation of each source and sink term to the change in $H_2SO_4$ concentration in Hyytiälä within the training data set. (D) Diurnal variation of each source and sink term to the change in $H_2SO_4$ concentration in Beijing within the training and testing data sets.

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
