# Peer review of "Sources and sinks driving sulphuric acid concentrations in contrasting environments: implications on proxy calculations"

_Atmospheric Chemistry and Physics, 2020_

## Referee Comment (RC1) · Santtu Mikkonen (Referee) · 8 Apr 2020

Review of Dada et al "*Sources and sinks driving sulphuric acid concentrations in contrasting environments: implications on proxy calculations*"

The manuscript addresses an important issue on predicting sulphuric acid concentrations when the measurements are not available. Especially finding an applicable proxy for night-time concentrations would be a significant improvement to existing literature. The manuscript introduces different variations of the proposed proxy and they seem to fit nicely on the measurements in selected locations. However, the procedure how the proxy variations were derived and the conditions where the measurements were made need to be described in more detail before the applicability of the proxies can be evaluated and I can recommend the manuscript for publication.

**Major comments:**

The proxies for individual campaigns were derived from the same data they are predicting, these proxies need to be verified on independent data before they can be generalized even on different conditions in the same sites. In addition, the data were collected from short periods, except for Hyytiälä, and it would be helpful if there would be some discussion on how representative the measurements are compared to annual level or long term seasonal averages of all variables in the sites. Bootstrap resampling is good method in the case where not so much comparable data are available but it is not enough for constructing a generalizable tool if the measurements are not representative.

Derivation of night-time proxies in Hyytiälä should be revisited. I would suggest calculating separate proxies for dark time without global radiation included, or similarly than in China, as the chemistry is different during the dark hours. The manuscript suggests that the night-time formation of sulphuric acid is mostly driven by Criegee intermediates and thus the coefficient $k_2$ in China was seen to be significantly higher than for daytime and that might be the case also in Hyytiälä.

**Specific comments:**

Page 2 line 76: proved->suggested

Page 2 lines 91-93: Bold statements, considering the comments in this revision regarding generalizability

Page 3, lines 102-104: Were all the measurements made on the same platform?

Page 3, lines 130-134: I have recently learnt that calibrating CI-APi-ToF is not an easy task (Talk by Ylisirniö et al. EAC2019). Were the instruments calibrated such that the results between sites are comparable and are the measured concentrations of realistic magnitude?

Page 4, lines 145-155: CS was reported in Hyytiälä with RH correction and in other sites no such correction is defined. The CS measures should be consistently defined if the results are being generalized.

Page 5, lines 183-184 and Figures S3-S7: Why Pearson correlation coefficients? The data are most probably not normally distributed and they contain outliers, which violate the basic assumptions of Pearson correlation.

Page 5, lines 203-209: How the sink term $k_3[H_2SO_4]^2$ is defined? It needs to be clarified here for usability of the proxy

Pages 5-6, Equations: Overall, the notation of the equations is somewhat confusing. First term is clear, does the second term refer similarly as the first one that it is $k_2$ times ozone concentration times Alkene concentration times $SO_2$ concentration? In addition, does $[H_2SO_4]$ in third term refer to sulphuric acid concertation or that the CS is calculated for sulphuric acid? Does in last term $[H_2SO_4]^2$ refer to squared concentration, and if yes, drawn from where? I suggest clarification of the equations.

Page 6, lines 242-249: It is not surprising to see that the Petäjä proxy had some difficulties, as it is constructed only with data from Hyytiälä. Already in Mikkonen et al. (2011) it was seen that the Petäjä proxy is not always working well outside of Hyytiälä. Thus, it would be interesting to see comparisons on proxy from Mikkonen et al., which has been shown to work in varying environments.

Page 6 lines 251-254: The predictor variables in the proxy contain high measurement uncertainty. Does the fminsearch procedure take that account?

Page 6 lines 254-257: I am happy to see uncertainty estimation for the coefficients made with bootstrap! Though some details on bootstrap procedure should be provided, e.g. how many resamples were drawn?

Page 6 lines 260-265: How does the AIC reflect the probability of over- or under-fitting in these analyses? As calculating log-likelihood for AIC might be sensitive for number of observations was it checked that the N was the same for all proxies in certain site? With multiple instruments in use, there might be gaps in data in different time points.

Page 7, line 273 and Figure 1: Are the numbers of data points the same in each subplot?

Figure 2 and related text in chapter 4.1: Do I read the figure correctly that the proxy values from 23-02 are missing? If this is due to missing global radiation, this could be corrected by the suggestion above to derive separate night-time proxy.

Page 7, line 308: "…proves the truthfulness…" is quite an overstatement

Figure 5: Why the scale is from $10^2$ when the data starts from $10^5$? Overall, the observed concentrations seem rather low for urban environment. Were the conditions somewhat unusual during the measurement campaign?

Page 9, lines 388-389: Clarify how the predicted fractions were drawn for table 2 and fig 9

Table 2: 27th percentile?

Figure 10: Global radiation distribution is missing. The basic statistics could also be given in (supplement) table. Sulphuric acid concentration in Megacity seems also low.

Page 10, lines 438-440: It is stated that the coefficients did not vary substantially, I might disagree. But regardless of that, did you try to pool the data from different sites an calculate a combined data proxy? Naturally with Equation 4 which could be calculated for all sites. Would this give a more generalizable proxy?

Discussion and suggestions section: It would be helpful to give here the direct equations for calculating the proxies in each site. It would probably increase the future use of the derived proxies. Equations could also be an appendix.

**References**

Mikkonen, S., et al. (2011) A statistical proxy for sulphuric acid concentration, Atmos. Chem. Phys. 11 11319-11334. doi:10.5194/acp-11-11319-2011

Ylisirniö et al. (2019) Solving the Mystery of Wildly Differing Results in Calibrating FIGAERO ToF-CIMS Desorption Temperature to Saturation Vapour Pressure. Proceedings of EAC2019, **O1_A6_M06**

---

## Referee Comment (RC2) · Anonymous Referee #2 · 13 May 2020

The manuscript, "Sources and sinks driving sulphuric acid concentrations in contrasting environments: implications on proxy calculations," by Dada et al. describes a new method for estimating gas phase H2SO4 concentrations using relatively common measurements. The development of these so-called "proxies" for H2SO4 is important as this species is often used in global models for simulating the timing and intensity of new particle formation events. Additional proxies are especially needed for representing regions that were not include in previous attempts (e.g., China) or during time periods that we not considered previously (e.g., nighttime). Thus, this manuscript is potentially valuable and is, in principle, worthy of publication in ACP. I do however, wish to point out a one main item and a few minor issues that I would like the authors to respond to

prior to recommending publication.

As a major concern: In the abstract of this manuscript and throughout the text the authors claim that the new proxy is "a more flexible and an important improvement of previous proxies." While that may be true, we only are provided a comparison to the previous proxy developed in a pristine boreal forest atmosphere (the Petaja proxy). Nowhere do the authors compare their new proxy to that developed by Mikkonen et al. First of all, this makes little sense as the Mikkonen model was developed for a broader range of conditions than the Petaja model. If there is a valid reason to disregard the Mikkonen model then the authors should state that, or else they should show model predictions from that on all relevant figures as they did with the Petaja model. Otherwise they should remove the statement that the model is an improvement over other proxies, as they are only comparing to one.

As minor issues: Line 27: Just to be slightly fussy with wording, H2SO4 is important in new particle formation for actually two reasons: it has low volatility and also has strong intramolecular bonding abilities. Merely mentioning low volatility misses qualities that make this compound special. Line 64: I suggest that the authors put a sentence or two here to state why it is important to develop a proxy for H2SO4. Many readers may be aware of the reason but it's a small thing to do and will be a great benefit to those who would otherwise be left wondering why so much effort is being placed in this. Line 75: I notice that Dr. Mikkonen is a reviewer of this article, so perhaps he will make this point (and I hope he also raises the concern that I express above). While the statement that his parameterization does not include condensation sink it technically correct, I believe that he considered this in his statistical analysis and found that condensation sink, or rather higher aerosol loading, is associated both with the source and sink of H2SO4, and that is the reason why on average it does not appear in the parameterization. If true then perhaps more accurate to state it this way rather than to leave the reader to conclude that this model overlooked the potential role of condensation sink. Line 86: I suggest you choose a better word than "goodness" Line 249: this reference to

the Petaja paper seems strange. Why wasn't standard referencing used is referring to Equation 7 in the text (e.g., on line 245)?

---

## Author Response (AR1)

Review of Dada et al. "Sources and sinks driving sulphuric acid concentrations in contrasting environments: implications on proxy calculations" by Santtu Mikkonen

The manuscript addresses an important issue on predicting sulphuric acid concentrations when the measurements are not available. Especially finding an applicable proxy for night-time concentrations would be a significant improvement to existing literature. The manuscript introduces different variations of the proposed proxy and they seem to fit nicely on the measurements in selected locations. However, the procedure how the proxy variations were derived and the conditions where the measurements were made need to be described in more detail before the applicability of the proxies can be evaluated and I can recommend the manuscript for publication.

**We thank Prof. Santtu Mikkonen for his valuable comments and suggestions, we think that these improve the applicability of the proxy and the overall quality of the study. We provided point-by-point answers in purple. Insertions to the text are in *Italics*. Line numbers refer to the ACPD version of the text.**

We thank Santtu again for his constructive comments. In order to address all comments and improve the quality of the manuscript the following developments have been done and their results were added to the manuscript.

To make the following sections straightforward and understandable we start by answering the specific comments 10 and 11 which are relevant to the method section prior to addressing the rest of the comments.

Page 6 lines 251-254: The predictor variables in the proxy contain high measurement uncertainty. Does the fminsearch procedure take that account?

Page 6 lines 254-257: I am happy to see uncertainty estimation for the coefficients made with bootstrap! Though some details on bootstrap procedure should be provided, e.g. how many resamples were drawn?

First of all, the measured data are now divided into independent training and testing data sets. The training sets are used for the derivation of the proxy equations and the testing data sets are used for testing the predictive power of the derived proxies. More details about those data sets are reported in both the main text and in more detail in the supplementary information.

The training sets are measured in Hyytiälä, Agia Marina, Budapest and Beijing. When used for deriving the proxy equation, 10 000 bootstrap resamples were introduced for each data set independently. Bootstrap resampling without disturbance generates extended data from the original data by randomly replacing an existing data point with another one from the same data set, resulting in different combinations of the original data set.

However, the reviewer is right, the fminsearch procedure does not take into account the measurement uncertainty of the predicting variables. Therefore, we included an estimate of error on each of the predictor variable, as well as on $H_2SO_4$, and included those when generating 10 000 random samples per variable per data point. This was done by scaling the entire time series of a variable by a scalar drawn from a uniform distribution of potential biases of the respective variable (arising for example from uncertainties in calibrations). We did not consider the precision error, since the accuracy error was considerably larger.

Let's take measured sulfuric acid concentration as an example. The measured concentration were accurate within a factor of 2. Therefore, while the temporal behavior of the variable was fairly certain, the entire time series might have been up to a factor too low or up to a factor too high. Therefore, we generated 10 000 concentrations by multiplying the original measured concentration by a uniform random array between the lower and upper bounds, which are 0.5 and 2 in the case of sulfuric acid. The same resampling method was applied for each of the other predictor variables as well as for $H_2SO_4$ independently, and the 10 000 possible combinations of the disturbed data sets were used to generate 10 000 different $k$ value combinations, therefore accounting for the errors in the variables. A median of these 10 000 $k$ values was then used to form one equation per location. Additionally, using the testing data sets, we explored whether predicting the concentration varies when we derive the concentration from the median k in the resulting equation, or when we derive it by using the 10 000 $k$ values and then taking the median concentration and the difference was negligible. A thorough description of the resampling method is now added to the supplementary information, in addition to the MATLAB code used. The introduction of the uncertainty to the predictor variables and $H_2SO_4$ widened the range of the $25^{th}$ and $75^{th}$ percentiles of the $k$ values (Table 1 – ACPD), while narrowing the contribution of each source and sink (Table 2 –ACPD).

The main text Line 251 now reads:

*The fitting coefficients were obtained by minimizing the sum of the squared logarithm of the ratio between the proxy values and measured sulphuric acid concentration using the method described by Lagarias et al. (1998), a build-in function fminsearch of MATLAB, giving the optimal values for the coefficients. The data were subject to 10 000 bootstrap resamples when getting each of the k values as a measure of accuracy in terms of bias, variance, confidence intervals, or prediction error (Efron and Tibshirani, 1994). We accounted for the systematic uncertainty in $H_2SO_4$ and predictor variables. For every bootstrap fit, we assumed both $H_2SO_4$ and all predictor variables to be affected by independent systematic errors between its lower and upper accuracy limits. More details on the bootstrap resampling method and uncertainty introduction can be found in the supplementary information. The $25^{th}$ percentile and $75^{th}$ percentiles of the coefficients are shown for all locations together with the median k values in Table 1. The median k values from the bootstrap resamples were used in the equations for deriving sulphuric acid concentrations at each site.*

The complementary section in the SI material now reads:

***Bootstrap resampling and sensitivity analyses***

*When deriving the proxy equation for each site, 10 000 bootstrap resamples were drawn for each data set independently. Bootstrap resampling without disturbance generates extended data from the original data by randomly replacing an existing data point with another one from the same data set, resulting in different combinations of variables from the original data set. We accounted for the systematic uncertainty in $H_2SO_4$ and predictor variables arising e.g. from calibration uncertainties. For every bootstrap fit, we assumed both $H_2SO_4$ and all predictor variables to be affected by independent systematic errors between the upper and lower bound of their independent uncertainty ranges. Since the uncertainty related to the measurement accuracy was much larger than the precision of the measurement, we only accounted for the uncertainty arising from accuracy. In practice, we scaled the entire time series of each variable by a random set of numbers drawn from a uniform distribution of possible measurement biases.*

*Accordingly, a factor of 2 uncertainty was introduced in the sulphuric acid concentration, a 20% uncertainty in the condensation sink measurement, and a 10% in each trace gas concentration and global radiation. In the case of sulphuric acid concentrations, which have a factor of 2 uncertainty, the actual concentration of sulphuric acid at a certain point in time could be anywhere between a factor of 2 lower and a factor of 2 higher. Therefore, for each sulphuric acid measurement, we generated 10 000 concentrations by multiplying the original measured concentration by a uniform random array between the lower and upper bounds, which are 0.5 and 2 in the case of sulphuric acid. The same resampling method ws applied for each other predictor variable independently, and the 10 000 possible combinations of the disturbed data sets were used to generate the fit and to derive the sulphuric acid proxy equation per site. A median of these 10 000 k value combinations which account for the error on the predictor variables was then used to form one equation per location. The MATLAB code used to generate the boot resamples is shown in Code 1.*

**Major Comments**

1. The proxies for individual campaigns were derived from the same data they are predicting, these proxies need to be verified on independent data before they can be generalized even on different conditions in the same sites.

2. In addition, the data were collected from short periods, except for Hyytiälä, and it would be helpful if there would be some discussion on how representative the measurements are compared to annual level or long term seasonal averages of all variables in the sites. Bootstrap resampling is good method in the case where not so much comparable data are available but it is not enough for constructing a generalizable tool if the measurements are not representative.

3. Derivation of night-time proxies in Hyytiälä should be revisited. I would suggest calculating separate proxies for dark time without global radiation included, or similarly than in China, as the chemistry is different during the dark hours. The manuscript suggests that the night-time formation of sulphuric acid is mostly driven by Criegee intermediates and thus the coefficient k2 in China was seen to be significantly higher than for daytime and that might be the case also in Hyytiälä.

**1. We explored the predictive power of our proxy by testing it on independent data sets.**

Each of the proxies of the boreal forest environment, rural background and mega city are tested for predictive power on independent data sets using extended data sets from the same location or using measurements from locations with similar characteristics (CS, trace gas concentrations – reference to Figure 10 in ACPD version). However, unfortunately our group has not performed any recent measurements in an urban location similar to the one in Budapest with a similar instrument or calibration, therefore for this specific site, we rely on bootstrap resampling only for accounting for variability in the predictor variables (Figures R1 – R15.

Overall, the modelled sulphuric acid concentrations correlated well (R = 0.7- Boreal; R = 0.45 – Rural and R = 0.83 – Megacity) with the measured sulphuric concentrations with a slope of ~1 for the testing data set except for the rural site, which could be attributed to the missing alkene source term resulting from the absence of alkene measurement in the Agia Marina data set. Additionally, we found that for all of the three testing data sets, the difference between the measured and modelled sulphuric acid concentrations was less than the error on the predication model itself for almost 70% of the data points. Note that the model prediction error was estimated as the interquartile range of the modelled $H_2SO_4$ concentration of a single point in time arising from the 10 000 different combinations in $k$ values (Figures R2, R6, R9 and R13).

1.1. Boreal environment:

The training data set used to develop the proxy equation was from August 18, 2016 to December 31, 2016 and from March 8, 2018 to February 28, 2019. For testing the predictive power of the proxy, we used an independent testing data set from January 1, 2017 to June 5, 2017 from the same location.

Hyytiälä proxy Equation 9:.

$$[H_2SO_4]_{boreal} = -\frac{CS}{2\ x\ (4.2\ x\ 10^{-9})} + \left[ \left( \frac{CS}{2\ x\ (4.2\ x\ 10^{-9})} \right)^2 + \frac{[SO_2]}{(4.2\ x\ 10^{-9})} (8.6\ x\ 10^{-9}\ x\ GlobRad + 6.1\ x\ 10^{-29}[O_3][Alkene]) \right]^{1/2}$$

The results from predicting sulphuric acid from the testing data sets using the above equation are shown in Figure R1 below, and the results from predicting sulphuric acid from 10 000 different $k$ value combinations specific to the site are shown in figure R2. Note that the 10 000 different $k$ value combinations refer to the 10 000 iterations performed on each time step including bootstrap resampling and accounting for predictor biases. Complementary error analyses to figure R2A are shown in figure R2B. The detailed method used to determine the $k$ value combinations from the training data set, as well as the one obtained from the equation above, are explained in details in the previous section. We also show the model prediction error which was estimated as the interquartile range of the modelled $H_2SO_4$ concentration of a single point in time arising from the uncertainty in $k$ values for each of the sites.

Moreover, we verified the four fits on the testing data set; i.e. the full Equation 2, the equation without the Stabilized Criegee Intermediates source (Equation 4), the equation without the cluster sink term (Equation 5) and the equation without neither the Stabilized Criegee Intermediates source nor the cluster sink term (Equation 6). We found that Fit 1 (Full equation) best defines the measured sulphuric acid concentration in comparison to the rest with a high correlation coefficient between the measured and the modelled data (R = 0.70) and a slope of 0.997 (Figure R3). The diurnal cycle is also nicely described by the Equation 4 which captures both nighttime and daytime (Figure R4).

[Figure]

*Figure R 1 Sulphuric acid concentrations modelled as a function of measured sulphuric acid at Hyytiälä SMEAR II station. The concentrations shown are 3-hour medians coinciding with the alkene measurements every three hours resulting in a total of 257 data points. The modelled concentrations are derived using equation 9. The colored data points refer to the modelled or predicted concentrations, the dashed blue line refers to the fit (log(y) = a.log(x)+b) of the aforementioned data points. The black squares are the median modelled concentrations in logarithmically spaced measured sulphuric acid bins and their lower and upper whistkers correspond to 25th and 75th percentiles of the predicted concentrations.*

(A)

(B)

[Figure]

*Figure R 2 (A) Sulphuric acid concentrations modelled as a function of measured sulphuric acid at Hyytiälä SMEAR II station. The concentrations shown are 3-hour medians coinciding with the alkene measurements every three hours resulting in a total of 257 data points. The modelled concentrations are the median derived using 10,000 k value combinations specific to the site. The colored data points refer to the modelled or predicted concentrations, the dashed blue line refers to the fit (log(y) = a.log(x)+b) of the aforementioned data points. The black squares are the median modelled concentrations in logarithmically spaced measured sulphuric acid bins and their lower and upper whistkers correspond to 25th and 75th percentiles of the predicted concentrations. (B) Cumulative distribution function of the model error weighted difference between measured and modeled $H_2SO_4$ concentration (using 257 data points).*

[Figure]

*Figure R 3 Sulphuric acid proxy concentration as a function of measured sulphuric acid observed at SMEAR II station, Hyytiälä Finland using the four different combinations of source and sink terms. The concentrations shown are 3-hour medians coinciding with the alkene measurements every three hours resulting in a total of 257 data points. In (A), the full Equation 2 is used, in (B) the equation without the Stabilized Criegee Intermediates source (Equation 4), in (C) the equation without the cluster sink term (Equation 5) and in (D) the equation without both the Stabilized Criegee Intermediates source and the cluster sink term (Equation 6). The 'Fit' refers to the fitting between the measured and the proxy calculated sulphuric acid concentration.*

[Figure]

*Figure R 4 The diurnal variation of sulphuric acid proxy concentrations using different fits and observed concentrations at SMEAR II in Hyytiälä, Finland. Median values are shown. Fits 1,2, 3 and 4 corresponds to the Equations 2, 4, 5, and 6, respectively. Petäjä fit shown is applied using the coefficients reported in (Petäjä et al., 2009)(Equation 7). Mikkonen fit shown is applied using the coefficients reported in Mikkonen et al. 2011 (Equation 8).*

**1.2. Rural location: Agia Marina Equation 10 (Glob Rad >= 50).**

$$[H_2SO_4]_{rural} = -\frac{CS}{2\ x\ (2.2\ x\ 10^{-9})} + \left[\left(\frac{CS}{2\ x\ (2.2\ x\ 10^{-9})}\right)^2 + \frac{[SO_2]}{(2.2\ x\ 10^{-9})}(9.7\ x\ 10^{-8}\ x\ GlobRad)\right]^{\frac{1}{2}}$$

An additional location 'Helsinki', representative of a semi-urban location was introduced for testing the predictive power of the rural proxy equation. Note that the rural equation was chosen over the urban equation, since the CS and SO$_2$ concentrations measured in Helsinki matched those in Agia Marina (rural location) rather than those in Budapest (urban location); see Figure 10 (ACPD). For testing the predictive power of the rural background site proxy (Equation 10), we used measurements from July 1, 2019 to July 16, 2019 during daytime (GlobRad >= 50 W/m$^2$). Results show that although the modelled sulphuric acid concentrations did not correlate as well as in other locations (R = 0.44), the bias could be attributed to the missing source (alkene) in the original equation as mentioned in the previous section. Indeed, looking at the binned data, we found that at within each concentration bin, the modelled sulphuric concentrations tend to span the 1:1 line. Actually, the discrepancy between the measured and the modelled concentration was smaller than the model prediction error (Figure R6). Note that the model prediction error was estimated as the interquartile range of the modelled H2SO4 concentration of a single point in time arising from the uncertainty in k values. For the rural background site, we also found that the diurnal cycle is better described when introducing the additional clustering sink term (Figure R7).

[Figure]

*Figure R 5 Sulphuric acid concentrations modelled as a function of measured sulphuric acid at Helsinki SMEAR III station. The concentrations shown are 1-hour medians resulting in a total of 416 data points. The modelled concentrations are derived using equation 10. The colored data points refer to the modelled or predicted concentrations, the dashed blue line refers to the fit (log(y) = a.log(x)+b) of the aforementioned data points. The black squares are the median modelled concentrations in logarithmically spaced measured sulphuric acid bins and their lower and upper whiskers correspond to 25th and 75th percentiles of the predicted concentrations.*

[Figure]

*Figure R 6 Sulphuric acid concentrations modelled as a function of measured sulphuric acid at Helsinki SMEAR III station. The concentrations shown are 1-hour medians resulting in a total of 416 data points. The modelled concentrations are the median derived using 10,000 k value combinations specific to the site. The colored data points refer to the modelled or predicted concentrations, the dashed blue line refers to the fit (log(y) = a.log(x)+b) of the aforementioned data points. The black squares are the median modelled concentrations in logarithmically spaced measured sulphuric acid bins and their lower and upper whistkers correspond to 25th and 75th percentiles of the predicted concentrations.(B) Cumulative distribution function of the model error weighted difference between measured and modeled H₂SO₄ concentration (using 416 data points).*

[Figure]

*Figure R 7 The diurnal variation of sulphuric acid proxy concentrations using different fits and observed concentrations at SMEAR III in Helsinki, Finland. Median values are shown. Fits 1,2, 3 and 4 corresponds to the Equations 2, 4, 5, and 6, respectively. Petäjä fit shown is applied using the coefficients reported in Petäjä et al. 2009 (Equation 7). Mikkonen fit shown is applied using the coefficients reported in Mikkonen et al. 2011 (Equation 8).*

**1.3.  Megacity: Beijing: Equation 12.**

$$[H_2SO_4]_{megacity} = -\frac{CS}{2\ x\ (7.0\ x\ 10^{-9})} + \left[\left(\frac{CS}{2\ x\ (7.0\ x\ 10^{-9})}\right)^2 + \frac{[SO_2]}{(7.0\ x\ 10^{-9})}(1.94\ x\ 10^{-8}\ x\ GlobRad + 1.44\ x\ 10^{-29}[O_3][Alkene])\right]^{1/2}$$

We applied the equation on an additional independent data set from the same location between September 8, 2019 and October 15, 2019. The results show that the modelled sulphuric acid concentrations correlated well (R = 0.84) with the measured sulphuric concentrations, with a slope of ~1.1 for the testing data set (Figure R8). Also for this site, we tested the four fits on the testing data set; i.e. the full Equation 2, the equation without the Stabilized Criegee Intermediates source (Equation 4), the equation without the cluster sink term (Equation 5) and the equation without neither the Stabilized Criegee Intermediates source nor the cluster sink term (Equation 6). We found that Fit 1 (Equation 4) best defines the measured sulphuric acid concentration in comparison to the rest of the equations. The results show a high correlation coefficient between the measured and the modelled data (R = 0.84) and a slope of 1.03 (Figure R10). The diurnal cycle is also nicely described by the Equation 4 which captures both nighttime and daytime (Figure R11).Similar to the boreal forest and rural site predictions, in Beijing, the discrepancy between the measured and the modelled concentration is also smaller than the model prediction error (Figure R9).

[Figure]

*Figure R 8 Sulphuric acid concentrations modelled as a function of measured sulphuric acid in Beijing. The concentrations shown are 1-hour medians resulting in a total of 268 data points. The modelled concentrations are derived using equation 12. The gray data points refer to the modelled or predicted concentrations, the dashed blue line refers to the fit (log(y) = a.log(x)+b) of the aforementioned data points. The black squares are the median modelled concentrations in logarithmically spaced measured sulphuric acid bins and their lower and upper whistkers correspond to 25th and 75th percentiles of the predicted concentrations.*

[Figure]

[Figure]

*Figure R 9 Sulphuric acid concentrations modelled as a function of measured sulphuric acid in Beijing. The concentrations shown are 1-hour medians resulting in a total of 263 data points. The modelled concentrations are the median derived using 10,000 k value combinations specific to the site. The gray data points refer to the modelled or predicted concentrations, the dashed blue line refers to the fit (log(y) = a.log(x)+b) of the aforementioned data points. The black squares are the median modelled concentrations in logarithmically spaced measured sulphuric acid bins and their lower and upper whiskers correspond to 25th and 75th percentiles of the predicted concentrations.(B) Cumulative distribution function of the model error weighted difference between measured and modeled $H_2SO_4$ concentration (using 263 data points).*

[Figure]

*Figure R 10 Sulphuric acid proxy concentration as a function of measured sulphuric acid observed at SMEAR II station, Hyytiälä Finland using the four different combinations of source and sink terms. The concentrations shown are 1-hour medians resulting in a total of 263 data points in each subplot. In (A), the full Equation 2 is used, in (B) the equation without the Stabilized Criegee Intermediates source (Equation 4), in (C) the equation without the cluster sink term (Equation 5) and in (D) the equation without both the Stabilized Criegee Intermediates source and the cluster sink term (Equation 6). The 'Fit' refers to the fitting between the measured and the proxy calculated sulphuric acid concentration (log(y) = a.log(x)+b).*

[Figure]

*Figure R 11 The diurnal variation of sulphuric acid proxy concentrations using different fits and observed concentrations at in Beijing, China for the testing data set. Median values are shown. Fits 1,2, 3 and 4 corresponds to the Equations 2, 4, 5, and 6, respectively. Petäjä fit shown is applied using the coefficients reported in Petäjä et al. 2009 (Equation 7). Mikkonen fit shown is applied using the coefficients reported in Mikkonen et al. 2011 (Equation 8).*

**1.4 Kilpilahti: Equation 10**

Finally, we did a very interesting test where we tested the predictive power of our developed proxy on a data set measured at an industrial area in close proximity to an oil refinery. Interestingly, the median CS at the location lies within the interquartile range of the CS measured in Hyytiälä and that measured in Agia Marina. The $SO_2$ concentrations at the measurement site were higher than in both Hyytiälä and Agia Marina, but smaller than the ones reported in Budapest. Additionally, we observed that alkene concentrations at Kilpilahti were within the range of those monitored in Hyytiälä, which is attributed to the green belt in the area (Sarnela et al., 2015). Accordingly, we tested the proxy equation 9 on the Kilpilahti data set. Our results showed that Equation 9 derived for Hyytiälä is able to predict the sulphuric acid concentrations in Kilpilahti with a high correlation coefficient (R= 0.74) (Figure R12). Similar to other locations, the Fit 1 (Equation 4) best describes the sources and sinks at the location (Figure R14). The discrepancy between the measured and the modelled concentration is smaller than the model prediction error for less than 50% of the data points only (Figure S13). This observation is consistent with the diurnal cycle (Figure R15). During certain mornings (4:00 – 8:00 LT), when the measured sulphuric concentrations were particularly high, the model was unable to predict the concentrations accurately. These high concentrations were attributed to air masses coming from the oil refinery (Sarnela et al., 2015). Indeed, our proxy was not able to explain these morning peaks using biogenic alkenes, however, in such an industrial area, anthropogenic sources could play a role in determining the magnitude of sulphuric acid concentrations. With the condensation sink being rather low (median ~0.005 s$^{-1}$), the impact of direct $H_2SO_4$ emissions cannot be ruled out either.

.

[Figure]

*Figure R 12 Sulphuric acid concentrations modelled as a function of measured sulphuric acid. The colored data points refer to the modelled (predicted) concentrations at Kilpilahti Finland, the dashed blue line refers to the fit (log(y) = a.log(x)+b) of the aforementioned data points. The black squares are the median modelled concentrations in logarithmically spaced measured sulphuric acid bins and their lower and upper whiskers correspond to 25th and 75th percentiles of the predicted concentrations. The concentrations shown are 1-hour medians resulting in 114 data points. The modelled concentrations are derived using equation 9.*

[Figure]

*Figure R 13 Sulphuric acid concentrations modelled as a function of measured sulphuric acid at Kilpilahti, Finland. The concentrations shown are 1-hour medians resulting in a total of 114 data points. The modelled concentrations are the median derived using 10,000 k value combinations specific to the the boreal forest location. The colored data points refer to the modelled or predicted concentrations, the dashed blue line refers to the fit (log(y) = a.log(x)+b) of the aforementioned data points. The black squares are the median modelled concentrations in logarithmically spaced measured sulphuric acid bins and their lower and upper whistkers correspond to 25th and 75th percentiles of the predicted concentrations. (B) Cumulative distribution function of the model error weighted difference between measured and modeled H2SO4 concentration (using 114 data points).*

[Figure]

*Figure R 14 Sulphuric acid proxy concentration as a function of measured sulphuric acid observed at Kilpilahti, oil refinary Finland using the four different combinations of source and sink terms derived from Hyytiälä. The concentrations shown are 1-hour medians resulting in a total of 114 data points in each subplot. In (A), the full Equation 2 is used, in (B) the equation without the Stabilized Criegee Intermediates source (Equation 4), in (C) the equation without the cluster sink term (Equation 5) and in (D) the equation without both the Stabilized Criegee Intermediates source and the cluster sink term (Equation 6). The 'Fit' refers to the fitting between the measured and the proxy calculated sulphuric acid concentration (log(y) = a.log(x)+b).*

[Figure]

*Figure R 15 The diurnal variation of sulphuric acid proxy concentrations observed concentrations at Kilpilahti, industrial area, Finland. Median values are shown. The modelled concentration is predicted using Equation 9 using the k values derived from Hyytiälä SMEAR II station.*

2. **Monthly variation of the sources and sinks in both Hyytiälä and Beijing**

Since our paper tackles mostly the sources and sinks of $H_2SO_4$ in various locations and not only aims at deriving a physical proxy and in order to assess the representative qualities of the data sets we used, we included monthly variation of the sources and sinks in both Hyytiälä and Beijing during which we have extended data sets which include nighttime calculations (Figure R16).

The text on Line 401 now reads:

*The Criegee intermediate term showed its importance mostly when global radiation is low, not only in nighttime but also during winter (Figure 11) in both Hyytiälä and Beijing.*

And on Line 414:

*The cluster term is found to contribute most during spring daytime in Hyytiälä (Figure 12 – A & C), which is the time window during which clustering and thus new particle formation events happen (Dada et al., 2018; Dada et al., 2017) The same is observed for Beijing, where the clustering term contributed up to 70% of the total sink terms during daytime (Figure 12-D).*

Additionally, we added a paragraph describing the representative nature of our data sets in comparison to the whole year for all site by comparing to available literature from each site.

The text on Line 141 now reads:

*Trace gases measured during the short campaign periods in Agia Marina and Budapest are representative of yearly concentrations in respective locations when compared to longer term measurements at the same site (Salma et al., 2016; Baalbaki, 2020, In Prep.).*

and on Line 155:

*Condensation sink values obtained during the short campaign periods in Agia Marina, Helsinki and Budapest are representative of yearly concentrations in respective locations when compared to longer term measurements at the same site (Salma et al., 2016; Baalbaki, 2020, In Prep.).*

(A)

[Figure]

Figure R 16 (A) Monthly variation of each source and sink term to the change in $H_2SO_4$ concentration in Hyytiälä during the period of the training data set 2016-2019 (excluding 2017). (B) Monthly variation of each source and sink term to the change in $H_2SO_4$ concentration in Beijing using a combined data set between January and December 2019. The data outside the training and testing data set has missing measured sulphuric acid concentration and proxy concentrations were used in obtaining this figure.

**3. Derivation of night-time proxies in Hyytiälä and Beijing**

We agree with the reviewer that the sources of the sulphuric acid may shift between day and night hours. Indeed, during dark hours, the Criegee intermediates' source is dominant. However, we think that extent of the contribution of each source term depends on the concentration of the precursor vapour rather than on the *k* itself, where k could be temperature dependent resulting in a difference between day and night. Nevertheless, we did the analysis for day and night separately. We compared the results from the separate (day and night) analysis to those from considering one equation as in Figure R 17.

First, we found that a better fit between the measured and training data set proxy concentrations is found when using one equation for daytime and nighttime than for daytime alone which has to do with the different points in time. Additionally, we found that the $k$ values derived from 10 000 iterations for all day, daytime and nighttime separately have distinct characteristics (Figure R19). First, $k_1$ values derived from all day, daytime alone or nighttime alone are within the range of each other. Interestingly, the $k_2$ values for daytime or nighttime alone are also similar, while when fitting one equation for daytime and nighttime together the $k_2$ values show different character. This means that separating the equation into day and night independently would depict the pattern of the predictor in this case the alkene term (Figure R20). The alkene term has a strong diurnal and seasonal cycle as shown in figure R20.

We performed the same analysis on the Beijing data set after we reassessed the Global Radiation data. In order to perform the 4 fits on any data set, the global radiation cannot be zero as otherwise Fit 2 fails completely. Therefore, in the case of Beijing we set the global radiation zero values into half the minimum observed radiation, which is assumed to be equivalent to the detection limit of the instrument ($GlobRad_{min} = 0.03$ W/m$^2$). After reassessing the global radiation data, we came to the same conclusion as for Hyytiälä, which is that one single equation for daytime and nighttime together is capable of explaining the sulphuric acid concentrations without Beijing biased to the diurnal or seasonal pattern of any of the predictor variables. The only obstacle was that when fitting one bulk equation for daytime and nighttime together unconstrained, the fit resulted in an unphysical $k_3$ value of the order of 0.01. In order to overcome this, we restricted the upper limit of the $k_3$ value to the median we get from fitting daytime data only. This assumption is acceptable since clustering is dominant during daytime. Indeed, when we then compared the daytime alone fits versus the ones from the bulk equation, we observed a better fit (Figure R21-R22). Additionally, different $k_1$ values for daytime and nighttime were obtained when fit separately, in general during the nighttime the global radiation is too low, and therefore has too low variability and therefore for this parameter the nighttime is poorly defined, which explains why the $k_1$ in this condition is an order of magnitude higher. When we fitted the data together, the $k_1$ matches the one from the daytime, which is not poorly defined. Therefore, also for Beijing we fitted the daytime and nighttime together (Figure R23). All in all, we think that introducing the predictive power of each of the equations, as suggested by the reviewer, was an excellent idea which helped in assessing whether using a bulk equation is enough for either location. Indeed, as shown in the previous section, for Hyytiälä the bulk proxy equation serves well in predicting both nighttime and daytime concentrations of sulphuric acids during the independent data set period. Similarly, obtaining the bulk equation from the spring time Beijing training data was able to predict both nighttime and daytime concentrations during summer and autumn in Beijing during the testing data set period.

However, in order to show the difference between daytime and nighttime in terms of sources or sinks, we decided to show diurnal contribution of those for both Hyytiälä in Beijing (Figure R 24-25). Similar to the observations from the monthly cycles, the diurnals show that when the global radiation is available the sulphuric acid formation pathway rather goes through the SO$_2$- OH mechanism. During dark hours, the Criegee pathway dominates the sulphuric acid source. Additionally, clustering is dominant during daytime hours. Please see insertions to the main text in the section above.

[Figure]

*Figure R 17 Sulphuric acid proxy concentration as a function of measured sulphuric acid. Observation at SMEAR II station, Hyytiälä Finland. The observed concentrations are measured 2016-2019 using CI-APi-ToF and are 3-hour medians resulting in a total of 1860 data points. In (A), the full Equation 2 is used, in (B) the equation without the Stabilized Criegee Intermediates source (Equation 4), in (C) the equation without the cluster sink term (Equation 5) and in (D) the equation without both the Stabilized Criegee Intermediates source and the cluster sink term (Equation 6). The 'Fit' refers to the fitting between the measured and the proxy calculated sulphuric acid concentration.*

[Figure]

*Figure R 18 Sulphuric acid proxy concentration as a function of measured sulphuric acid **during daytime** (GlobRad >= 50 W/m²). Observation at SMEAR II station, Hyytiälä Finland. The observed concentrations are measured 2016-2019 using CI-APi-ToF and are 3-hour medians for daytime data resulting in a total of 921 data points. In (A), the full Equation 2 is used, in (B) the equation without the Stabilized Criegee Intermediates source (Equation 4), in (C) the equation without the cluster sink term (Equation 5) and in (D) the equation without both the Stabilized Criegee Intermediates source and the cluster sink term (Equation 6). The 'Fit' refers to the fitting between the measured and the proxy calculated sulphuric acid concentration.*

[Figure]

*Figure R 19 Histograms showing the occurence of k values derived from 10,000 disturbed booststrap resampling runs when fitting a full-day proxy denoted by 'All' and colored in blue, a daytime proxy denoted by 'Daytime' and colored in green, and a nighttime proxy denoted by 'Nighttime' and colored in grey.*

[Figure]

*Figure R 20 Temporal variation in the median monoterpene concentration in Hyytiälä 2016- 2019. Observation at SMEAR II station, Hyytiälä Finland. The observed concentrations are measured 2016-2019 using PTR-ToF, see also Perakyla et al. (2014).*

[Figure]

*Figure R 21 (A) Sulphuric acid proxy concentration as a function of measured sulphuric acid. Observation at Beijing, China. The observed concentrations of the training data set are measured in 2019 using CI-APi-ToF and are 1-hour medians resulting in a total of 877 data points. In (A), the full Equation 2 is used, in (B) the equation without the Stabilized Criegee Intermediates source (Equation 4), in (C) the equation without the cluster sink term (Equation 5) and in (D) the equation without both the Stabilized Criegee Intermediates source and the cluster sink term (Equation 6). Coefficients shown on top of the subplots relate to the daytime values. The 'Fit' refers to the fitting between the measured and the proxy calculated sulphuric acid concentration (log(y) = a.log(x)+b). Note that the upper limit of the cluster term k value is limited to the same value as the daytime value to avoid getting unphysical values which were observed ($k_3$ = 0.01) in case no limit on the k value is added.*

[Figure]

*Figure R 22 (A) Sulphuric acid proxy concentration as a function of measured sulphuric acid. Observation at Beijing, China during daytime GlobRad >= 50 W/m². The observed concentrations of the training data set are measured in 2019 using CI-APi-ToF and are 1-hour medians resulting in a total of 415 data points. In (A), the full Equation 2 is used, in (B) the equation without the Stabilized Criegee Intermediates source (Equation 4), in (C) the equation without the cluster sink term (Equation 5) and in (D) the equation without both the Stabilized Criegee Intermediates source and the cluster sink term (Equation 6). Coefficients shown on top of the subplots relate to the daytime values. The 'Fit' refers to the fitting between the measured and the proxy calculated sulphuric acid concentration (log(y) = a.log(x)+b).*

[Figure]

*Figure R 23 Histograms showing the occurrence of k values derived from 10,000 disturbed bootstrap resampling runs when fitting a full-day proxy denoted by 'All' and colored in blue, a daytime proxy denoted by 'Daytime' and colored in green, and a nighttime proxy denoted by 'Nighttime' and colored in grey in Beijing. Note that the $k_3$ values are not shown since they are similar to the daytime values due to limiting the $k_3$ to the upper limit of the daytime $k_3$ value.*

.

[Figure]

*Figure R 24 Diurnal variation of each source and sink term to the change in $H_2SO_4$ concentration in Hyytiälä within the training data set.*

[Figure]

*Figure R 25 Diurnal variation of each source and sink term to the change in $H_2SO_4$ concentration in Beijing within the training data set.*

Specific comments:

Point-by-point replies to the specific comments are added below.

1.  Page 2 line 76: proved->suggested

Modified.

2.  Page 2 lines 91-93: Bold statements, considering the comments in this revision regarding generalizability

Modified.

*In order to evaluate our hypothesized sources and sinks and derive the proxy equations, we utilize measurements from four different locations: (1) Hyytiälä, Finland, (2) Agia Marina, Cyprus, (3) Budapest, Hungary and (4) Beijing, China, representing a semi-pristine boreal forest environment, rural environment in the Mediterranean area, urban environment and heavily polluted megacity, respectively. To evaluate the predictive power of the derived proxies, the equations are further tested on independent data sets. We further compare the coefficients of production and losses in each environment in order to understand the prevailing mechanism of the $H_2SO_4$ budget in each of the studied environments. As a result of this investigation, a well-defined sulphuric acid concentration can be derived for multiple areas around the world and even extended in time during times when it was not measured (such as: gap filling, forecast, prediction, estimation, etc.).*

3.  Page 3, lines 102-104: Were all the measurements made on the same platform?

Measurements of different variables within the same location are performed at the same platform except for Hyytiälä and Helsinki. We added details related to the measurement platforms of every variable to section 2.2.

4.  Page 3, lines 130-134: I have recently learnt that calibrating CI-APi-ToF is not an easy task (Talk by Ylisirniö et al. EAC2019). Were the instruments calibrated such that the results between sites are comparable and are the measured concentrations of realistic magnitude?

We agree that different organic compounds calibrations are still mystery (Talk by Ylisirniö et al. EAC2019), however, calibrations of sulphuric acid are straightforward and robust. The instruments in all four locations were calibrated in a similar way using the method presented by (Kurten et al., 2012) and the results are comparable.

We added the following to the Line 134:

*In all locations, the CI-APi-ToF instruments were calibrated in a similar way prior to the campaign using the method presented by Kurten et al. (2012) to ensure the results are comparable.*

5.  Page 4, lines 145-155: CS was reported in Hyytiälä with RH correction and in other sites no such correction is defined. The CS measures should be consistently defined if the results are being generalized.

We agree with the reviewer that including a hygroscopic growth correction for only the boreal forest results in a discrepancy when inter-comparing. Therefore, we reassessed the fits for the boreal forest location using condensation sink values calculated in the same way as in the rest of the studied locations. The results of this fit would be suitable for comparing the sources and sinks in various locations. We replaced the equation and related k values in the main text with those reassessed, see figure R26.

However, we think that the boreal forest environment has been studied thoroughly over the years and it is ideal to use the best data we have and all the information we could. In case any of the readers is interested in calculating a sulphuric acid proxy from Hyytiälä, we recommend that they use the equation which includes corrected CS for hygroscropic growth.

In fact, we found that the fit with the hygroscopic correction is better than that without this correction. See figure R17 (no correction) in comparison to figure R26 (with correction).

The results and equations are added into the supplementary information and the related text in main text. Line 302 now reads:

*Furthermore, we derived an additional proxy equation using CS corrected for hygroscopic growth (Laakso et al., 2004) to be used when calculating a more robust proxy for Hyytiälä. The details, equation and results are shown in the supplementary information (Figure S10-S12).*

[Figure]

*Figure R 26 Sulphuric acid proxy concentration as a function of measured sulphuric acid. Observation at SMEAR II station, Hyytiälä Finland **with CS corrected for hygroscopic growth**. The observed concentrations are measured 2016-2019 using CI-APi-ToF and are 3-hour medians resulting in a total of 1594 data points. In (A), the full Equation 2 is used, in (B) the equation without the Stabilized Criegee Intermediates source (Equation 4), in (C) the equation without the cluster sink term (Equation 5) and in (D) the equation without both the Stabilized Criegee Intermediates source and the cluster sink term (Equation 6). The 'Fit' refers to the fitting between the measured and the proxy calculated sulphuric acid concentration(log(y)=a.log(x)+b).*

[Figure]

*Figure R 27 The diurnal variation of sulphuric acid proxy concentrations using different fits and observed concentrations at SMEAR II in Hyytiälä, Finland. Median values are shown. Fits 1,2, 3 and 4 corresponds to the Equations 2, 4, 5, and 6, respectively. Petäjä fit shown is applied using the coefficients reported in Petäjä et al. 2009 (Equation 7). Mikkonen fit shown is applied using the coefficients reported in Mikkonen et al. 2011 (Equation 8).*

6.  Page 5, lines 183-184 and Figures S3-S7: Why Pearson correlation coefficients? The data are most probably not normally distributed and they contain outliers, which violate the basic assumptions of Pearson correlation.

The reviewer is right. We used the scatter plots between the variables to decide which coefficient we should use. We replaced the Pearson with a Spearman coefficients in Figures S2-S6.

7.  Page 5, lines 203-209: How the sink term $k_3[H2SO4]^2$ is defined? It needs to be clarified here for usability of the proxy.
8.  Pages 5-6, Equations: Overall, the notation of the equations is somewhat confusing. First term is clear, does the second term refer similarly as the first one that it is $k_2$ times ozone concentration times Alkene concentration times SO2 concentration? In addition, does [H2SO4] in third term refer to sulphuric acid concertation or that the CS is calculated for sulphuric acid? Does in last term [H2SO4] 2 refer to squared concentration, and if yes, drawn from where? I suggest clarification of the equations.

As per the suggestion of the two previous comments a clarification has been added to the text to explain the 3$^{rd}$ and 4$^{th}$ terms of the Equation 1

$$\frac{d[H_2SO_4]}{dt} = k_0[OH][SO_2] + k_2[O_3][Alkene][SO_2] - CS[H_2SO_4] - k_3[H_2SO_4]^2 \qquad (1)$$

The text on line 201 now read:

*The third term in Equation 1 represents the loss of $H_2SO_4$ into pre-existing aerosol particles, known as condensation sink (CS) and is calculated by multiplying the CS calculated for sulphuric acid with the concentration of sulphuric acid monomer. The fourth term in Equation 1 is defined as the square of sulphuric acid concentration multiplied by clustering coefficient $k_3$. The square of sulphuric acid represents the collision of two sulphuric acid monomers forming a sulphuric acid dimer, which was found to be the first step of atmospheric cluster formation (Yao et al., 2018). Therefore, this term takes into account the additional loss of $H_2SO_4$ due to cluster formation not included in the term containing CS. This is necessary because CS is only inferred from size-distribution measurements at maximum down to 1.5 nm, i.e. not containing any cluster concentrations and hence losses onto these clusters. This term is written in the form of sulphuric acid dimer production, which seems to be the first step of cluster formation once stabilized by bases (Kulmala et al., 2013; Almeida et al., 2013; Yao et al., 2018).*

9.  Page 6, lines 242-249: It is not surprising to see that the Petäjä proxy had some difficulties, as it is constructed only with data from Hyytiälä. Already in Mikkonen et al. (2011) it was seen that the Petäjä proxy is not always working well outside of Hyytiälä. Thus, it would be interesting to see comparisons on proxy from Mikkonen et al., which has been shown to work in varying environments.

We compared our proxies with Mikkonen et al. 2011 in all 4 locations, and added the diurnal Mikkonen plot to the main text (Figures 2,4, 6 and 8) while the scatter plots between measured sulphuric acid concentrations and both of Petäjä and Mikkonen proxies during daytime (GlobRad >= 50 W/m$^2$) in Figures S13 and S14, respectively.

Hyytiälä                                      Agia Marina

[Figure]

Budapest                                      Beijing

[Figure]

*Figure R 28 The diurnal variation of sulphuric acid proxy concentrations using different fits and observed concentrations. Median values are shown. Fits 1,2, 3 and 4 corresponds to the Equations 2, 4, 5, and 6, respectively. Petäjä fit shown is applied using the coefficients reported in Petäjä et al. 2009 (Equation 7) and Mikkonen et al. 2011 (Equation 8).*

**Hyytiälä**

**Agia Marina**

**Budapest**

**Beijing**

*Figure R 29 Scatter plot showing the correlation between measured sulphuric acid and the sulphuric acid concentrations derived from the Petäjä et al. 2009 proxy at the 4 locations during daytime (GlobRad >= 50 W/m2): Hyytiälä, Agia Marina, Budapest and Beijing.*

[Figure]

*Figure R 30 Scatter plot showing the correlation between measured sulphuric acid and the sulphuric acid concentrations derived from the Mikkonen et al. 2011 proxy at the 4 locations during daytime (GlobRad >= 50 W/m²): Hyytiälä, Agia Marina, Budapest and Beijing.*

10. Page 6 lines 251-254: The predictor variables in the proxy contain high measurement uncertainty. Does the fminsearch procedure take that account?

11. Page 6 lines 254-257: I am happy to see uncertainty estimation for the coefficients made with bootstrap! Though some details on bootstrap procedure should be provided, e.g. how many resamples were drawn?

Answers to question 10 and 11 are added to the beginning of this document.

12. Page 6 lines 260-265: How does the AIC reflect the probability of over- or under-fitting in these analyses? As calculating log-likelihood for AIC might be sensitive for number of observations was it checked that the N was the same for all proxies in certain site? With multiple instruments in use, there might be gaps in data indifferent time points.

The reviewer is right that the AIC criterion is sensitive or even driven by the N. In order to avoid the bias due to number of observation points per fit, we selected the data points when all variables are available simultaneously. We also add a Table S4 which shows the parameters included in deriving the AIC in each site. See also next comment.

13. Page 7, line 273 and Figure 1: Are the numbers of data points the same in each subplot?

For each location separately, all the subplots contain the same number of points. Although it might be possible to include more points in the panels where no alkene term is included, yet for comparability reasons, especially for the AIC we kept a constant number of data points per subplot. The number of points to each of the subplot for all 4 locations is shown in the corresponding figure caption. A table S4 describing the statistics included in the AIC calculation such as the number of points, correlation coefficients, slope .. etc. is added to the supplementary information.

14. Figure 2 and related text in chapter 4.1: Do I read the figure correctly that the proxy values from 23-02 are missing? If this is due to missing global radiation, this could be corrected by the suggestion above to derive separate night-time proxy.

There is no missing data except that the PTR measurements for alkenes are every 3 hours. We are sorry for the typo in the figure 1 caption. Now it is corrected.

15. Page 7, line 308: "…proves the truthfulness…" is quite an overstatement

We agree with the reviewer that using the same data set for deriving and predicating is not a valid method for a proxy derivation. Besides adding a complete section on the predictive powers of the derived proxies, we modified the above sentence into:

*The correlation between the measured and proxy concentration of $H_2SO_4$ was 0.88 (96 data points) which shows that the chosen predictors were able to explain the measured sulphuric acid concentration largely (Figure 3).*

16. Figure 5: Why the scale is from $10^2$ when the data starts from $10^5$? Overall, the observed concentrations seem rather low for urban environment. Were the conditions somewhat unusual during the measurement campaign?

The figure is fixed. Concerning the overall concentrations, we do not think that there were any unusual conditions. The measured concentrations are within the range of observations between Hyytiälä and Beijing. We added a time series of the measured $H_2SO_4$ in Budapest in the supplementary information (Figure S1) to help show the variation in the $H_2SO_4$ concentrations upon changes in meteorology.

17. Page 9, lines 388-389: Clarify how the predicted fractions were drawn for table 2 and fig 9

Line 389 now reads:

*The contribution of the various source and sink terms to the change of $H_2SO_4$ concentrations are determined using Equation 2. The median derived $k_1$, $k_2$ and $k_3$ values, together with the measured $H_2SO_4$, CS, trace gases and GlobRad per site, were used to calculate each of the terms. Source term 1 refers to $k_1$ x GlobRad x [$SO_2$], source term 2 refers to $k_2$ x [$O_3$] x [Alkene] x [$SO_2$], sink term 3 refers to $k_3$ x [$H_2SO_4$]$^2$ and sink term 4 refers to CS x [$H_2SO_4$]. The contribution of each term is then calculated as the median or percentiles of the normalized term to the sum of all terms.*

18. Table 2: 27th percentile?

This was a typo, we changed it to 75$^{th}$.

19. Figure 10: Global radiation distribution is missing. The basic statistics could also be given in (supplement) table. Sulphuric acid concentration in Megacity seems also low.

Global radiation distribution and a table of basic statistics was added to the supplementary information.

20. Page 10, lines 438-440: It is stated that the coefficients did not vary substantially, I might disagree. But regardless of that, did you try to pool the data from different sites an calculate a combined data proxy? Naturally with Equation 4 which could be calculated for all sites. Would this give a more generalizable proxy?

We agree with the reviewer that unifying the parametrization with the aim of coming up with 1 equation would be nice. In this sense, we unified the day and night time equations wherever possible and present now unified equations each for Beijing and Hyytiälä. These equations perform well in explaining the diurnal variability at the respective site. Unifying wasn't possible for the Cyprus and Budapest datasets because of missing alkene data. Merging Hyytiälä and Beijing to come up with a single proxies would require accounting for different alkene mixes (boreal forest dominated by biogenic VOCs, Beijing strongly impacted by anthropogenic VOCs). And yes, we revisited our k values, illustrated in Figure R31, the $k_2$ related to the sulphuric acid formation through Criegee intermediates is clearly different at both locations. Additionally, with the different sizes of data sets from each of the locations, when we tried to assess one parametrization using Equation 4, as suggested with the reviewer, the fit was bias to the Hyytiälä data which has the highest contribution. Therefore, we opt here not to further unify, yet agree with the reviewer that such efforts should be targeted in future results together with distinguishing further chemical processes such as the contribution of different VOC classes.

[Figure]

*Figure R 31 Histogram showing the distribution of k2 values from 10,000 iterations in both Hyytiälä and Beijing.*

21. Discussion and suggestions section: It would be helpful to give here the direct equations for calculating the proxies in each site. It would probably increase the future use of the derived proxies. Equations could also be an appendix.

The equations 9-12 are added to Table 1.

References

[revised manuscript text omitted]

Review of Dada et al "Sources and sinks driving sulphuric acid concentrations in contrasting environments: implications on proxy calculations" by Anonymous Referee

The manuscript, "Sources and sinks driving sulphuric acid concentrations in contrasting environments: implications on proxy calculations," by Dada et al. describes a new method for estimating gas phase H2SO4 concentrations using relatively common measurements. The development of these so-called "proxies" for H2SO4 is important as this species is often used in global models for simulating the timing and intensity of new particle formation events. Additional proxies are especially needed for representing regions that were not include in previous attempts (e.g., China) or during time periods that we not considered previously (e.g., nighttime). Thus, this manuscript is potentially valuable and is, in principle, worthy of publication in ACP. I do however, wish to point out a one main item and a few minor issues that I would like the authors to respond to prior to recommending publication.

We thank the reviewer for their valuable comments and suggestions, we think that these help improve the presentation of the proxy and the overall quality of the study. We provided point-by-point answers in purple. Insertions to the text are in *Italics*. Line numbers refer to the old version of the ACPD version of the text.

As a major concern: In the abstract of this manuscript and throughout the text the authors claim that the new proxy is "a more flexible and an important improvement of previous proxies." While that may be true, we only are provided a comparison to the previous proxy developed in a pristine boreal forest atmosphere (the Petaja proxy). Nowhere do the authors compare their new proxy to that developed by Mikkonen et al. First of all, this makes little sense as the Mikkonen model was developed for a broader range of conditions than the Petaja model. If there is a valid reason to disregard the Mikkonen model then the authors should state that, or else they should show model predictions from that on all relevant figures as they did with the Petaja model. Otherwise they should remove the statement that the model is an improvement over other proxies, as they are only comparing to one.

We agree with the reviewer that it is rather crucial to compare to Mikkonen et al. as it has been developed for several locations including a broad range of conditions. However, since our proxy includes periods that we have not considered previously (e.g., nighttime), we still think that it is an improvement over previous proxies.

We compared our proxies with Mikkonen et al. 2011 in all 4 locations, and added the diurnal Mikkonen plot to the main text (Figures 2,4, 6 and 8) while the scatter plots between measured sulphuric acid concentrations and both of Petäjä and Mikkonen proxies during daytime (GlobRad >= 50 W/m$^2$) in Figures S13 and S14, respectively.

[Figure]

Hyytiälä

Agia Marina

Budapest

Beijing

*Figure R 1 The diurnal variation of sulphuric acid proxy concentrations using different fits and observed concentrations. Median values are shown. Fits 1,2, 3 and 4 corresponds to the Equations 2, 4, 5, and 6, respectively. Petäjä fit shown is applied using the coefficients reported in Petäjä et al. 2009 (Equation 7) and Mikkonen et al. 2011 (Equation 8).*

[Figure]

*Figure R 2 Scatter plot showing the correlation between measured sulphuric acid and the sulphuric acid concentrations derived from the Petäjä et al. 2009 proxy at the 4 locations during daytime (GlobRad >= 50 W/m²): Hyytiälä, Agia Marina, Budapest and Beijing.*

[Figure]

*Figure R 3 Scatter plot showing the correlation between measured sulphuric acid and the sulphuric acid concentrations derived from the Mikkonen et al. 2011 proxy at the 4 locations during daytime (GlobRad >= 50 W/m²): Hyytiälä, Agia Marina, Budapest and Beijing.*

As minor issues:

1. Line 27: Just to be slightly fussy with wording, H2SO4 is important in new particle formation for actually two reasons: it has low volatility and also has strong intramolecular bonding abilities. Merely mentioning low volatility misses qualities that make this compound special.

We agree with the reviewer that $H_2SO_4$ is distinct for its strong hydrogen bonding ability which makes it possible to interact with other species and is found to be important for the first step of cluster formation. We have modified the relevant sentence on Line 58 to the following:

*Sulphuric acid ($H_2SO_4$), which has a very low saturation vapor pressure and strong hydrogen bonding capability (Zhang et al., 2011), has been found to be the major precursor of atmospheric NPF (Weber et al., 1996; Kulmala et al., 2004; Sihto et al., 2006; Sipilä et al., 2010; Erupe et al., 2011; Lehtipalo et al., 2018; Ma et al., 2019) and is often used in global models for simulating the occurrence and intensity of new particle formation events*.

2. Line 64: I suggest that the authors put a sentence or two here to state why it is important to develop a proxy for H2SO4. Many readers may be aware of the reason but it's a small thing to do and will be a great benefit to those who would otherwise be left wondering why so much effort is being placed in this.

We added the following sentences as per recommendation from the reviewer:

Line 60: *Sulphuric acid ($H_2SO_4$), which has a very low saturation vapor pressure, has been found to be the major precursor of atmospheric NPF (Weber et al., 1996; Kulmala et al., 2004; Sihto et al., 2006; Sipilä et al., 2010; Erupe et al., 2011; Lehtipalo et al., 2018; Ma et al., 2019) and is often used in global models for simulating the occurrence and intensity of new particle formation events (Dunne et al., 2016).*

and to Line 80:

*Besides the abovementioned-previously-developed proxies, an additional proxy is still needed for representing nighttime periods which were not considered previously.*

3. Line 75: I notice that Dr. Mikkonen is a reviewer of this article, so perhaps he will make this point (and I hope he also raises the concern that I express above). While the statement that his parameterization does not include condensation sink it technically correct, I believe that he considered this in his statistical analysis and found that condensation sink, or rather higher aerosol loading, is associated both with the source and sink of H2SO4, and that is the reason why on average it does not appear in the parameterization. If true then perhaps more accurate to state it this way rather than to leave the reader to conclude that this model overlooked the potential role of condensation sink.

We did not intend to say that Mikkonen et al. (2011) have overlooked the potential role of condensation sink, we have however referred to their sentence in the abstract copied below.

Sentence from Dada et al. 2020: "Proxies developed by Mikkonen et al. (2011) suggested that the sulphuric acid concentration depends mostly on the available radiation and $SO_2$ concentration, with little influence of CS."

Sentence from Mikkonen et al. 2011: "Interestingly, the role of the condensation sink in the proxy was only minor, since similarly accurate proxies could be constructed with global solar radiation and SO2 concentration alone."

4. Line 86: I suggest you choose a better word than "goodness"

We modified the sentence to the following:

*In order to evaluate the accuracy of the our hypothesized sources and sinks and derive the proxy equations goodness of our new proxy, we utilize measurements from four different locations: (1) Hyytiälä, Finland, (2) Agia Marina, Cyprus, (3) Budapest, Hungary and (4) Beijing, China, representing a semi-pristine boreal forest environment, rural environment in the Mediterranean area, urban environment and heavily polluted megacity, respectively. To evaluate the predictive power of the derived proxies, the equations are further tested on independent data sets.*

5. Line 249: this reference to Petaja paper seems strange. Why wasn't standard referencing used is referring to Equation 7 in the text (e.g., on line 245)?

We thank the reviewer for noticing; we modified the related text to the following:

[revised manuscript text omitted]

**1. Reaction rate constant from Mikkonen et al. 2011**
Derivation of the temperature dependent reaction rate constant ($k$) used in calculating the
Mikkonen proxy from our data sets:

$$k \ (cm^3 \ molec^{-1} \ s^{-1}) = \frac{A \cdot k_3}{(A+k_3)} \ x \exp\left\{ k_5 \left[ 1 + \log_{10}\left(\frac{A}{k_3}\right)^2 \right]^{-1} \right\} \qquad (S1)$$

$$A = k_1 \cdot [M] \cdot \left(\frac{300}{k}\right)^{k_2} \qquad (S2)$$

$$[M] = 0.101 \cdot (1.381 \ x \ 10^{-23} \ T)^{-1} \qquad (S3)$$

M is the density of the air in molec cm$^{-3}$, $k_1 = 4\times10^{-31}$, $k_2 = 3.3$, $k_3 = 2\times10^{-12}$ and $k_5 = -0.8$.
$k$ given in Equation (S1) is scaled by multiplying it with $10^{12}$ as described in more detail in Mikkonen
et al. (2011).
**2. Bootstrap resampling and sensitivity analyses**
When deriving the proxy equation for each site, 10 000 bootstrap resamples were drawn for each data
set independently. Bootstrap resampling without disturbance generates extended data from the
original data by randomly replacing an existing data point with another one from the same data set,
resulting in different combinations of variables from the original data set. We accounted for the
systematic uncertainty in H$_2$SO$_4$ and predictor variables arising e.g. from calibration uncertainties.
For every bootstrap fit, we assumed both H$_2$SO$_4$ and all predictor variables to be affected by
independent systematic errors between the upper and lower bound of their independent uncertainty
ranges. Since the uncertainty related to the measurement accuracy was much larger than the precision
of the measurement, we only accounted for the uncertainty arising from accuracy. In practice, we
scaled the entire time series of each variable by a random set of numbers drawn from a uniform
distribution of possible measurement biases.
Accordingly, a factor of 2 uncertainty was introduced in the sulphuric acid concentration, a 20%
uncertainty in the condensation sink measurement, and a 10% in each trace gas concentration and
global radiation. In the case of sulphuric acid concentrations, which have a factor of 2 uncertainty,
the actual concentration of sulphuric acid at a certain point in time could be anywhere between a
factor of 2 lower and a factor of 2 higher. Therefore, for each sulphuric acid measurement, we
generated 10 000 concentrations by multiplying the original measured concentration by a uniform
random array between the lower and upper bounds, which are 0.5 and 2 in the case of sulphuric acid.
The same resampling method ws applied for each other predictor variable independently, and the 10
000 possible combinations of the disturbed data sets were used to generate the fit and to derive the
sulphuric acid proxy equation per site. A median of these 10 000 k value combinations which account
for the error on the predictor variables was then used to form one equation per location. The MATLAB
code used to generate the boot resamples is shown in Code 1.

*Code  1. MATLAB code used to generate the boot resamples and obtain the fitting coefficients ($k_1$, $k_2$*

*and $k_3$) using Equation 3.*

```matlab
%% Derive k values for sulfuric acid proxy concentration using Dada et al. 2020 equation
% fitCoeff(1) = k1
% fitCoeff(2) = k2
% fitCoeff(3) = k3

data = [CS, SO2, O3, Alkene, GlobRad]; %CS in s-1, SO2,O3,Alkene in cm-3, GlobRad in W/m2
H2SO4; %measured sulfuric acid in cm-3

% Create the fitting function according to Equation 3

Y_fit = @(fitCoeff,data) (-1).*(data(:,1)./(2*fitCoeff(3))) + ...
    sqrt((data(:,1)./(2*fitCoeff(3))).^2 + data(:,2)./(fitCoeff(3)).*...
    (fitCoeff(1).*data(:,5) + fitCoeff(2).*data(:,3).*data(:,4)));

% Obtain the fitting coefficients were obtained by minimizing the sum of the squared logarithm
%of the ratio between the proxy values and measured sulphuric acid concentration sum_squared_error = @(fit_coeff) sum((log10(H2SO4 ./ (Y_fit(fit_coeff,data)))).^2);

%introduce bootstrap resampling fit_index = 10000; %number of bootstrap resampling

[~,bootsam] = bootstrp(fit_index,sum_squared_error,data); %bootstrap resampling

%introduce uncertainty estimates on the measured predictor varianbles
%create an array of random floating-point numbers that are drawn from a
%uniform distribution in the open interval between the lower and upper bound of accuracy

% 20% uncertainty on condensation sink
a = log10(1/1.2); %lower bound accuracy
b = log10(1.2); %upper bound accuracy
r_CS = 10.^((b-a).*rand(fit_index,1)+a);

% factor of 2 uncertainty on H2SO4 measurement
a = log10(0.5);
b = log10(2);
r_SA = 10.^((b-a).*rand(fit_index,1) + a);

% 10% uncertainty on trace gases and global radiation
a = log10(1/1.1);
b =log10(1.1);
r_SO2 = 10.^((b-a).*rand(fit_index,1) + a); %SO2

a = log10(1/1.1);
b =log10(1.1);
r_O3 = 10.^((b-a).*rand(fit_index,1) + a); %O3

a = log10(1/1.1);
b =log10(1.1);
r_MT = 10.^((b-a).*rand(fit_index,1) + a); %Alkenes a = log10(1/1.1);
b =log10(1.1);
r_GR = 10.^((b-a).*rand(fit_index,1) + a); %GlobRadiation

%
k_all=[];
for i =1:fit_index

    %create bootstrapped data disturbed with uncertainty on predictor variables
data_boot = [data(bootsam(:,i),1)*r_CS(i),data(bootsam(:,i),2)*r_SO2(i),...
    data(bootsam(:,i),3)*r_O3(i), data(bootsam(:,i),4)*r_MT(i),...
    data(bootsam(:,i),5)*r_GR(i)];
H2SO4_boot=H2SO4(bootsam(:,i),:)*r_SA(i);

% Obtain the fitting coefficients for the bootstrap resamples sum_squared_error = @(fit_coeff) sum((log10(H2SO4_boot ./ (Y_fit(fit_coeff,data_boot)))).^2);

% Assume initial values for the fitting parameters:
k0 = [1e-8, 1e-27,1e-9];

% Use built-in MATLAB function fminsearch to find the fitting parameters;
% the best fit parameters are in output into variable k:
[k, SSE] = fminsearch (sum_squared_error, k0, options); k_all = [k_all;k(:,:)];
end
```

*Table S 1 Summary of measurement locations and instrumentation used for deriving the $H_2SO_4$ proxy*

*(training data sets).*

| Location | Type | Measurement Period | Particle size distribution instrument | Trace Gases | Radiation |
|---|---|---|---|---|---|
| Hyytiälä, Finland | Boreal | August 18, 2016 to December 31,  2016 and March 8, 2018 to February 28, 2019 | Twin  DMPS (Ground level). | $SO_2$ and $O_3$ are monitored using two Thermo Environmental Instruments (models 43i-TLE, 49i, respectively), at 16.8 m above ground level. | [1]Global radiation was measured with Middleton solar SK08 pyranometer until August 24, 2017 and after that with Middleton solar EQ08-S pyranometer. at 16.8 m. |
| Agia Marina, Cyprus[2] | Rural background | February 22 and March 3, 2018 | 2-20 nm using Airel NAIS and 20-800 nm using TSI SMPS | $SO_2$ and $O_3$ are monitored using Ecotech Instruments (9850 and 9810, respectively) | Campbell Scientific weather station |
| Budapest, Hungary | Urban | March 21 and April 17, 2018 | 6-1000 nm using flow-switching type DMPS | $SO_2$ is measured using UV fluorescence (Ysselbach 43C) | Global radiation was measured by an SMP3 pyranometer (Kipp and Zonnen, The Netherlands) |
| Beijing, China | MegaCity | March 15, 2019 – June 15, 2019 | 3 – 800 nm PSD system ~12 m above ground level. | $SO_2$ and $O_3$ are monitored using two Thermo Environmental Instruments (models 43i-TLE, 49i, respectively), ~12 m above ground.. | [3]Global radiation was measured using CMP11 pyranometer (Kipp and Zonnen, Delft, Netherlands) at ~ 15 m above ground level. |
* * *
[1] UVB radiation was measured with Solar SL 501A pyranometer.

[2] All variables are measured at the same height.

[3] UVB radiation was measured using a UVS-B-T radiometer (Kipp and Zonnen, Delft, Netherlands).

*Table S 2 Summary of measurement locations and instrumentation used for verifying the predictive*
*power of the derived proxies (testing data sets).*

| Location | Type | Measurement Period | Particle size distribution instrument | Trace Gases | Radiation |
|---|---|---|---|---|---|
| Hyytiälä, Finland | Boreal | January 1, 2017 – June 5, 2017 | Twin – DMPS(Ground level). | $SO_2$ and $O_3$ are monitored using two Thermo Environmental Instruments (models 43i-TLE, 49i, respectively). | Global radiation was measured with Middleton solar EQ08-S pyranometer. |
| Helsinki, Finland | Semi-urban | July 1, 2019 – July 16, 2019 | Twin DMPS at ground level | $SO_2$ was measured using UV-flurescence (Horiba APSA 360) at 31 m above ground | Global radiation was monitored Kipp and Zonen CNR1 at 31 m above ground level |
| Beijing, China | MegaCity | September 8, 2019 – October 15, 2019 | 3 – 800 nm PSD system ~12 m above ground | $SO_2$ and $O_3$ are monitored using two Thermo Environmental Instruments (models 43i-TLE, 49i, respectively) ~ 12 m above ground.. | Global radiation was measured using CMP11 pyranometer (Kipp and Zonnen, Delft, Netherlands) at ~ 15 m above ground level. |
| Kilpilahti, Finland | Industrial Area | June 07, 2012 – June 29, 2012 | 6 to 1000 nm DMPS. | $SO_2$ was monitored usingThermo Scientific ™ Model 43i $SO_2$ Analyser | Acquired from SMEAR III station. |

Table S 3 Summary of basic statistics of measurements of condensation sink, trace gases and global radiation at all locations and time periods included in this study. For Hyytiälä, Beijing and Kilpilahti we use all day time window, for Agia Marina, Budapest and Helsinki we use daytime statistics (GlobRad > 50 W/m$^2$).

| Location | | Hyytiälä, Finland | Hyytiälä, Finland | Agia Marina, Cyprus | Helsinki, Finland | Budapest Hungary | Beijing, China | Beijing, China | Kilpilahti, Finland |
|---|---|---|---|---|---|---|---|---|---|
| *Type* | | Boreal | Boreal | Rural | Semi-urban | Urban | MegaCity | MegaCity | Industrial Area |
| *Measurement Period* | | August 18 - December 31, 2016 March 8 - February 28, 2019 | January 1, 2017 – June 5, 2017 | February 22 - March 3, 2018 | July 1 – July 16, 2019 | March 21 - April 17, 2018 | March 15, 2019 – June 15, 2019 | September 8, 2019 – October 15, 2019 | June 07, 2012 – June 29, 2012 |
| *CS ($10^{-3}$ s$^{-1}$)* | mean | 4.48 | 2.88 | 4.43 | 3.38 | 11.74 | 24.20 | 23.22 | 5.25 |
| | median | 3.83 | 2.18 | 3.63 | 3.13 | 10.92 | 22.83 | 22.60 | 4.91 |
| | 5$^{th}$ percentile | 0.85 | 0.74 | 1.37 | 1.25 | 5.03 | 7.60 | 5.14 | 2.61 |
| | 95$^{th}$ percentile | 12.43 | 8.78 | 9.58 | 6.47 | 21.52 | 44.58 | 44.34 | 8.81 |
| | sd | 3.89 | 2.42 | 2.55 | 1.60 | 5.37 | 11.86 | 11.82 | 2.11 |
| *SO$_2$ ($10^{10}$ cm$^{-3}$)* | mean | 0.31 | 0.30 | 0.70 | 1.30 | 6.02 | 4.70 | 2.43 | 6.65 |
| | median | 0.12 | 0.16 | 0.46 | 0.87 | 5.45 | 3.49 | 1.35 | 2.98 |
| | 5$^{th}$ percentile | 0.03 | 0.01 | 0.17 | 0.13 | 3.35 | 0.26 | 0.13 | 0.99 |
| | 95$^{th}$ percentile | 1.24 | 1.01 | 1.96 | 2.19 | 12.42 | 13.71 | 8.47 | 26.00 |
| | sd | 0.54 | 0.47 | 0.65 | 3.19 | 2.54 | 4.59 | 3.56 | 11.46 |
| *O$_3$ ($10^{10}$ cm$^{-3}$)* | mean | 83.59 | 95.08 | | | | 105.63 | 116.10 | 161.36 |
| | median | 80.27 | 97.10 | | | | 95.66 | 102.53 | 178.15 |
| | 5$^{th}$ percentile | 41.09 | 65.42 | | | | 5.23 | 3.24 | 24.81 |
| | 95$^{th}$ percentile | 134.85 | 118.42 | | | | 238.26 | 260.97 | 234.37 |
| | sd | 28.52 | 16.80 | | | | 72.22 | 80.99 | 62.92 |
| *Alkene ($10^{10}$ cm$^{-3}$)* | mean | 0.92 | 0.32 | | | | 14.33 | 11.98 | 2.27 |
| | median | 0.39 | 0.15 | | | | 12.29 | 11.91 | 0.72 |
| | 5$^{th}$ percentile | 0.05 | 0.02 | | | | 1.91 | 2.55 | 0.11 |

| | | | | | | | | |
|---|---|---|---|---|---|---|---|---|
| | 95th percentile | 3.54 | 0.85 | | | | 34.40 | 19.51 | 10.20 |
| | sd | 2.03 | 0.98 | | | | 9.68 | 4.96 | 3.38 |
| Global Radiation $(W.m^{-2})$ | mean | 149.25 | 93.06 | 283.71 | 353.67 | 322.90 | 243.72 | 221.27 | 307.86 |
| | median | 47.53 | 23.17 | 272.48 | 270.60 | 300.56 | 54.27 | 52.97 | 252.64 |
| | 5th percentile | 0.47 | 0.36 | 67.92 | 61.59 | 70.64 | 0.02 | 0.02 | 0.06 |
| | 95th percentile | 636.60 | 378.50 | 548.90 | 837.27 | 697.42 | 840.95 | 730.83 | 768.84 |
| | sd | 205.18 | 137.32 | 155.33 | 254.08 | 200.36 | 308.33 | 273.10 | 280.05 |
| $H_2SO_4$ $(10^6\ cm^{-3})$ | mean | 0.73 | 0.55 | 2.76 | 3.82 | 1.54 | 2.94 | 3.45 | 10.59 |
| | median | 0.28 | 0.18 | 1.81 | 2.55 | 1.02 | 1.61 | 2.00 | 3.19 |
| | 5th percentile | 0.02 | 0.02 | 0.17 | 0.41 | 0.23 | 0.37 | 0.37 | 0.19 |
| | 95th percentile | 2.55 | 2.01 | 8.22 | 11.71 | 4.76 | 8.63 | 10.98 | 37.08 |
| | sd | 1.40 | 1.06 | 3.06 | 4.57 | 1.77 | 3.00 | 3.74 | 28.25 |

*Table S 4 Statistical parameters included in deriving the Aikake Information Criterion. Equation*
*number refers to the number in the main text, N is the sample size (number of points), X is the number*
*of coefficients (number of k values) and SSE is the sum of squared estimate of errors. AIC is calculated*
*as AIC = 2X + N ln(SSE). The quantity exp((AIC$_{min}$ − AIC$_i$)/2) describes the probability that the ith*
*model minimizes the information loss. For example, Equation 5 in Hyytiälä is 5.62E-8 times as*
*probable as the Equation 6 to minimize the information loss.*

| | | | | | |
|---|---|---|---|---|---|
| **Hyytiälä Eq. 9** | Equation number | 6 | 5 | 4 | 2 |
| | number of coefficients | 3 | 2 | 2 | 1 |
| | N | 1860 | 1860 | 1860 | 1860 |
| | R | 0.84 | 0.74 | 0.82 | 0.70 |
| | Slope | 0.80 | 0.78 | 0.96 | 1.84 |
| | SSE | 1.89E+02 | 3.00E+02 | 2.88E+02 | 1.17E+03 |
| | AIC | 4.24E+03 | 4.61E+03 | 4.58E+03 | 5.71E+03 |
| | exp((AIC$_{min}$ – AIC$_i$)/2) | 1 | 5.62E-81 | 5.09E-74 | 0 |
| **Cyprus Eq. 10** | Equation number | 6 | 5 | 4 | 2 |
| | number of coefficients | 3 | 2 | 2 | 1 |
| | N | | 96 | | 96 |
| | R | | 0.88 | | 0.80 |
| | Slope | | 0.53 | | 0.67 |
| | SSE | | 2.02 | | 5.22 |
| | AIC | | 33.30 | | 69.86 |
| | exp((AIC$_{min}$ – AIC$_i$)/2) | | 1 | | 1.15E-08 |
| **Budapest Eq. 11** | Equation number | 6 | 5 | 4 | 2 |
| | number of coefficients | 3 | 2 | 2 | 1 |
| | N | | 263 | | 263 |
| | R | | 0.59 | | 0.49 |
| | Slope | | 0.47 | | 0.95 |
| | SSE | | 10.73 | | 30.10 |
| | AIC | | 275.06 | | 389.85 |
| | exp((AIC$_{min}$ – AIC$_i$)/2) | | 1 | | 1.19E-25 |
| **Beijing Eq. 12** | Equation number | 6 | 5 | 4 | 2 |
| | number of coefficients | 3 | 2 | 2 | 1 |
| | n | 877 | 877 | 877 | 877 |
| | R | 0.72 | 0.89 | 0.70 | 0.90 |
| | Slope | 1.69 | 3.16 | 2.11 | 5.23 |
| | SSE | 189.72 | 318.04 | 275.05 | 769.09 |
| | AIC | 2003.90 | 2198.67 | 2143.37 | 2532.00 |
| | exp((AIC$_{min}$ – AIC$_i$)/2) | 1 | 2.57E-85 | 2.69E-61 | 4.4E-230 |

[Figure]

*Figure S 1 SO₂ and measured H₂SO₄ concentrations in Budapest showing the change in concentration*
*due to changes in meteorology mid-campaign.*

[Figure]

*Figure S 2 Effect of hygroscopic growth correction on condensation sink calculation in the boreal*
*forest. Figure S 3*
Figure S 2

[Figure]

[Figure]

*Figure S 2 Spearman's correlation coefficients matrix between variables involved in $H_2SO_4$*

*formation and loss at the Hyytiälä station (Global Radiation > 0 $W/m^2$). CS represents condensation*

*sink in $s^{-1}$. $SO_2$, $O_3$ and MT (monoterpenes) in molecules/$cm^{-3}$. GlobRad is global radiation in $W/m^2$.*

*$H_2SO_4$ is measured sulphuric acid in molecules/$cm^{-3}$. The color bar represents the Spearman's*

*correlation coefficient. In (A) the condensation sink is not corrected for hygroscopic growth, while*

*in (B) the condensation sink is corrected for hygroscopic growth using the parametrization given by*

*Laakso et al. (2004).*

[Figure]

[Figure]

*Figure S 3  Spearman's correlation coefficients matrix of variables involved in $H_2SO_4$*
*formation and loss at the Agia Marina station (Global Radiation > 50 W/m$^2$). CS represents*
*condensation sink in s$^{-1}$. SO$_2$ is in molecules/cm$^{-3}$. GlobRad is global radiation in W/m$^2$. H$_2$SO$_4$ is*
*measured sulphuric acid in molecules/cm$^{-3}$.The color bar represents the Spearman's correlation*
*coefficient.*

[Figure]

[Figure]

*Figure S 4* *Spearman's correlation coefficients matrix of variables involved in $H_2SO_4$ formation and loss at the Budapest station (Global Radiation > 50 $W/m^2$). CS represents condensation sink in $s^{-1}$. $SO_2$ in molecules/$cm^{-3}$. GlobRad is global radiation in $W/m^2$. $H_2SO_4$ is measured sulphuric acid in molecules/$cm^{-3}$. The color bar represents the Spearman's correlation coefficient.*

[Figure]

[Figure]

*Figure S 5* *Spearman's correlation coefficients matrix between variables involved in $H_2SO_4$*
*formation and loss at the Beijing station* *. CS represents*
*condensation sink in $s^{-1}$. $SO_2$, $O_3$ and*  *Alkenes (Anthropogenic volatile organic compounds)*
*in molecules/cm³. GlobRad is global radiation in $W/m^2$. $H_2SO_4$ is measured sulphuric acid in*
*molecules/cm³. The color bar represents the Spearman's correlation coefficient.*

[Figure]

[Figure]

Figure S 6 *Spearman's* correlation coefficients matrix between variables involved in $H_2SO_4$ formation and loss at the Beijing station . CS represents condensation sink in $s^{-1}$. $SO_2$, $O_3$ and  *Alkenes* (Anthropogenic volatile organic compounds) in molecules/cm$^{-3}$. GlobRad is global radiation in $W/m^2$. $H_2SO_4$ is measured sulphuric acid in molecules/cm$^{-3}$. *The color bar represents the Spearman's correlation coefficient. In (A) the daytime correlation coefficients are shown (Global radiation >= 50 W/m²) and in (B) the nighttime correlation coefficents are shown (Global radiation < 50 W/m²).*

[Figure]

*Figure S 7 Comparison between Global radiation and UVB in Hyytiälä. Hourly medians are shown. The total number of data points in the plot is 2306.*

[Figure]

*Figure S 8 Comparison between Global radiation and UVB in Beijing. Hourly medians are shown. The total number of data points in the plot is 7106.*

[Figure]

*Figure S 9 Evaluation of the goodness of the fit using the Akaike information criterion (AIC) (McElreath, 2018). Number of parameters refers to the number of variables in each equation used. For example, Equation 2 uses four parameters which are the two sources (Radiation and sCI) and the two sinks (CS and cluster formation).*

[Figure]

*Figure S 10 Effect of hygroscopic growth correction on condensation sink calculation in the boreal forest. The solid line is the 1:1 line and the dashed lines are the 2:1 lines.*

[Figure]

*Figure S 11 Sulphuric acid proxy concentration as a function of measured sulphuric acid. Observation at SMEAR II station, Hyytiälä Finland **with CS corrected for hygroscopic growth**. The observed concentrations are measured 2016-2019 using CI-APi-ToF and are 3-hour medians resulting in a total of 1594 data points. In (A), the full Equation 2 is used, in (B) the equation without the Stabilized Criegee Intermediates source (Equation 4), in (C) the equation without the cluster sink term (Equation 5) and in (D) the equation without both the Stabilized Criegee Intermediates source and the cluster sink term (Equation 6). The 'Fit' refers to the fitting between the measured and the proxy calculated sulphuric acid concentration(log(y)=a.log(x)+b).*

[Figure]

*Figure S 12 The diurnal variation of sulphuric acid proxy concentrations using different fits and observed concentrations at SMEAR II in Hyytiälä, Finland. Median values are shown. Fits 1,2, 3 and 4 corresponds to the Equations 2, 4, 5, and 6, respectively. Petäjä fit shown is applied using the coefficients reported in Petäjä et al. 2009 (Equation 7). Mikkonen fit shown is applied using the coefficients reported in Mikkonen et al. 2011 (Equation 8).*

[Figure]

*Figure S 8 Comparison between Global radiation and UVB in Hyytiälä. Hourly medians are shown. The total number of data points in the plot is 2306.*

[Figure]

*Figure S 9 Comparison between Global radiation and UVB in Beijing. Hourly medians are shown.*
*The total number of data points in the plot is 7106.*

[Figure]

*Figure S 10 Evaluation of the goodness of the fit using the Akaike information criterion (AIC)*
*(McElreath, 2018). Number of parameters refers to the number of variables in each equation used.*
*For example, Equation 2 uses four parameters which are the two sources (Radiation and sCI) and*
*the two sinks (CS and cluster formation).*

[Figure]

*Figure S 11 Sulphuric acid proxy concentration as a function of measured sulphuric acid. observation at BUCT station, Beijing, China for day and nighttime combined. The observed concentrations are measured 2018-2019 using CI-APi-ToF and are 1-hour medians resulting in a total of 902 data points. In (A), the full Equation 2 is used, in (B) the equation without the Stabilized Criegee Intermediates source (Equation 4), in (C) the equation without the cluster sink term (Equation 5) and in (D) the equation without both the Criegee Intermediates source and the cluster sink term (Equation 6).*

[Figure]

Figure S 12 The diurnal variation of sulphuric acid proxy concentrations using different fits and observed concentrations at Beijing China. Median values are shown. Fits 1,2, 3 and 4 corresponds to the Equations 2, 4, 5, and 6, respectively. Petäjä fit shown is applied using the coefficients reported in Petäjä et al. 2009.

*Hyytiälä*

*Agia Marina*

[Figure]

*Figure S 13  Scatter plot showing the correlation between measured sulphuric acid and the sulphuric acid concentrations derived from the Petäjä et al. 2009 proxy at the 4 locations during daytime(GlobRad >= 50 W/m²): Hyytiälä, Agia Marina, Budapest and Beijing.*

[Figure]

*Figure S 14  Scatter plot showing the correlation between measured sulphuric acid and the sulphuric acid concentrations derived from the Mikkonen et al. 2011 proxy at the 4 locations during daytime (GlobRad >= 50 W/m²): Hyytiälä, Agia Marina, Budapest and Beijing.*

[Figure]

Figure S 15 Daytime data (GlobRad > 50 W/m²) condensation sink, SO₂,GlobRad and H₂SO₄ concentrations in diffrerent environements. The concentrations are displayed as violin plots which are a combination of boxplot and a kernel distribution function on each side of the boxplots. The white circles define the median of the distribution and the edges on the inner grey boxes refer to the 25th and 75th percentiles respectively.

(A)   (B)

[Figure]

*Figure S 16 (A) Sulphuric acid concentrations modelled as a function of measured sulphuric acid at Hyytiälä SMEAR II station. The concentrations shown are 3-hour medians coinciding with the alkene measurements every three hours resulting in a total of 257 data points. The modelled concentrations are the median derived using 10,000 k value combinations specific to the site. The colored data points refer to the modelled or predicted concentrations, the dashed blue line refers to the fit ($\log(y) = a.\log(x)+b$) of the aforementioned data points. The black squares are the median modelled concentrations in logarithmically spaced measured sulphuric acid bins and their lower and upper whistkers correspond to 25$^{th}$ and 75$^{th}$ percentiles of the predicted concentrations. (B) Cumulative distribution function of the model error weighted difference between measured and modeled $H_2SO_4$ concentration (using 257 data points).*

[Figure]

none
*Figure S 17 Sulphuric acid proxy concentration as a function of measured sulphuric acid observed at SMEAR II station, Hyytiälä Finland using the four different combinations of source and sink terms. The concentrations shown are 3-hour medians coinciding with the alkene measurements every three hours resulting in a total of 257 data points. In (A), the full Equation 2 is used, in (B) the equation without the Stabilized Criegee Intermediates source (Equation 4), in (C) the equation without the*

*cluster sink term (Equation 5) and in (D) the equation without both the Stabilized Criegee*
*Intermediates source and the cluster sink term (Equation 6). The 'Fit' refers to the fitting between*
*the measured and the proxy calculated sulphuric acid concentration.*

[Figure]

*Figure S 18 The diurnal variation of sulphuric acid proxy concentrations using different fits and*
*observed concentrations at SMEAR II in Hyytiälä, Finland. Median values are shown. Fits 1,2, 3 and*
*4 corresponds to the Equations 2, 4, 5, and 6, respectively. Petäjä fit shown is applied using the*
*coefficients reported in (Petäjä et al., 2009)(Equation 7). Mikkonen fit shown is applied using the*
*coefficients reported in Mikkonen et al. 2011 (Equation 8).*

[Figure]

*Figure S 19 Sulphuric acid concentrations modelled as a function of measured sulphuric acid at*
*Helsinki SMEAR III station. The concentrations shown are 1-hour medians resulting in a total of 416*
*data points. The modelled concentrations are the median derived using 10,000 k value combinations*
*specific to the site. The colored data points refer to the modelled or predicted concentrations, the*
*dashed blue line refers to the fit (log(y) = a.log(x)+b) of the aforementioned data points. The black*
*squares are the median modelled concentrations in logarithmically spaced measured sulphuric acid*
*bins and their lower and upper whistkers correspond to 25th and 75th percentiles of the predicted*
*concentrations. (B) Cumulative distribution function of the model error weighted difference between*
*measured and modeled H₂SO₄ concentration (using 416 data points).*

[Figure]

*Figure S 20 The diurnal variation of sulphuric acid proxy concentrations using different fits and*
*observed concentrations at SMEAR III in Helsinki, Finland. Median values are shown. Fits 1,2, 3*
*and 4 corresponds to the Equations 2, 4, 5, and 6, respectively. Petäjä fit shown is applied using the*
*coefficients reported in Petäjä et al. 2009 (Equation 7). Mikkonen fit shown is applied using the*
*coefficients reported in Mikkonen et al. 2011 (Equation 8).*

(A) (B)

[Figure]

*Figure S 21 Sulphuric acid concentrations modelled as a function of measured sulphuric acid in*
*Beijing. The concentrations shown are 1-hour medians resulting in a total of 263 data points. The*
*modelled concentrations are the median derived using 10,000 k value combinations specific to the*
*site. The gray data points refer to the modelled or predicted concentrations, the dashed blue line*
*refers to the fit (log(y) = a.log(x)+b) of the aforementioned data points. The black squares are the*
*median modelled concentrations in logarithmically spaced measured sulphuric acid bins and their*
*lower and upper whiskers correspond to 25th and 75th percentiles of the predicted concentrations.(B)*
*Cumulative distribution function of the model error weighted difference between measured and*
*modeled $H_2SO_4$ concentration (using 268 data points). $H_2SO_4$ concentration relative to the measured*
*$H_2SO_4$ concentration (using 268 data points).*

[Figure]

*Figure S 22 Sulphuric acid proxy concentration as a function of measured sulphuric acid observed*
*at Beijing, China for the testing data set using the four different combinations of source and sink*
*terms. The concentrations shown are 1-hour medians resulting in a total of 268 data points in each*
*subplot. In (A), the full Equation 2 is used, in (B) the equation without the Stabilized Criegee*
*Intermediates source (Equation 4), in (C) the equation without the cluster sink term (Equation 5) and*
*in (D) the equation without both the Stabilized Criegee Intermediates source and the cluster sink term*

*(Equation 6). The 'Fit' refers to the fitting between the measured and the proxy calculated sulphuric*
*acid concentration (log(y) = a.log(x)+b).*

[Figure]

*Figure S 23 The diurnal variation of sulphuric acid proxy concentrations using different fits and*
*observed concentrations at in Beijing, China for the testing data set. Median values are shown. Fits*
*1,2, 3 and 4 corresponds to the Equations 2, 4, 5, and 6, respectively. Petäjä fit shown is applied*
*using the coefficients reported in Petäjä et al. 2009 (Equation 7). Mikkonen fit shown is applied using*
*the coefficients reported in Mikkonen et al. 2011 (Equation 8).*

[Figure]

Figure S 24 *Sulphuric acid concentrations modelled as a function of measured sulphuric acid at Kilpilahti, Finland. The concentrations shown are 1-hour medians resulting in a total of 114 data points. The modelled concentrations are the median derived using 10,000 k value combinations specific to the the boreal forest location. The colored data points refer to the modelled or predicted concentrations, the dashed blue line refers to the fit (log(y) = a.log(x)+b) of the aforementioned data points. The black squares are the median modelled concentrations in logarithmically spaced measured sulphuric acid bins and their lower and upper whistkers correspond to 25th and 75th percentiles of the predicted concentrations. (B) Cumulative distribution function of the model error weighted difference between measured and modeled $H_2SO_4$ concentration (using 114 data points).*

[Figure]

Figure S 25 *Sulphuric acid proxy concentration as a function of measured sulphuric acid observed at Kilpilahti, oil refinary Finland using the four different combinations of source and sink terms derived from Hyytiälä. The concentrations shown are 1-hour medians resulting in a total of 114 data points in each subplot. In (A), the full Equation 2 is used, in (B) the equation without the Stabilized Criegee Intermediates source (Equation 4), in (C) the equation without the cluster sink term (Equation 5) and in (D) the equation without both the Stabilized Criegee Intermediates source and*

*the cluster sink term (Equation 6). The 'Fit' refers to the fitting between the measured and the proxy*
*calculated sulphuric acid concentration (log(y) = a.log(x)+b).*

[Figure]

*Figure S 26 The diurnal variation of sulphuric acid proxy concentrations observed concentrations at*
*Kilpilahti, industrial area, Finland. Median values are shown. The modelled concentration is*
*predicted using Equation 9 using the k values derived from Hyytiälä SMEAR II station.*